# GUIDED AUTOREGRESSIVE DIFFUSION MODELS WITH APPLICATIONS TO PDE SIMULATION

**Federico Bergamin**[1*], **Cristiana Diaconu**[2*], **Aliaksandra Shysheya**[2*], **Paris Perdikaris**[3],
**José Miguel Hernández-Lobato**[2], **Richard E. Turner**[2,3], **Emile Mathieu**[2]
[1]Technical University of Denmark, [2]University of Cambridge, [3]Microsoft Research AI4Science
`fedbe@dtu.dk`,`{cdd43,as2975,ret26,jmh233,ebm32}@cam.ac.uk`,
`paperdikaris@microsoft.com`

## ABSTRACT

Solving partial differential equations (PDEs) is of crucial importance in science and engineering. Yet numerical solvers necessitate high space-time resolution which in turn leads to heavy computational cost. Often applications require solving the same PDE many times, only changing initial conditions or parameters. In this setting, data-driven machine learning methods have shown great promise, a principle advantage being the ability to simultaneously train at coarse resolutions and produce fast PDE solutions. In this work we introduce the Guided AutoRegressive Diffusion model (GUARD), which is trained over short segments from PDE trajectories and a posteriori sampled by conditioning over (1) some initial state to tackle *forecasting* and/or over (2) some sparse space-time observations for *data assimilation* purposes. We empirically demonstrate the ability of such a sampling procedure to generate accurate predictions of long PDE trajectories.

## 1 INTRODUCTION

Partial differential equations (PDEs) are ubiquitous as they are a powerful mathematical framework for modelling physical phenomena. Solving PDEs relies on numerical methods like finite element methods (Liu et al., 2022) which require very small space and time discretisation to obtain accurate and reliable solutions resulting in significant computational resources. There has been a rising interest in leveraging deep learning methods to learn neural approximations of PDE solutions (Raissi et al., 2017). However, since dynamics arising from PDEs can be particularly complex or chaotic (Hyman & Nicolaenko, 1986; Kevrekidis et al., 1990), a major challenge is to predict long trajectories whilst preserving accuracy and stability (Kochkov et al., 2021) since ambiguities accumulate. Building on prior work, we propose to address these issues taking a *probabilistic* treatment of PDE dynamics (Yang & Sommer, 2023; Cachay et al., 2023) by leveraging diffusion models (Sohl-Dickstein et al., 2015; Ho et al., 2020; Song et al., 2021). In particular, autoregressive amortised diffusion style of models, which work by additionally feeding previous states to the score network, are currently state of the art for long rollouts (Lippe et al., 2023; Kohl et al., 2023). However, (a) prior work has found that higher Markov order harms performance, and (b) amortisation does not straightforwardly handle data assimilation.

In this work we show how a diffusion model trained on short segments of joint states—coined GUARD— can solve both of these limitations. Our contributions are the following: **(i)** We introduce a joint diffusion model which can be used autoregressively for sampling variable length trajectories, by iteratively generating states conditioned on past ones via reconstruction guidance (Song et al., 2022). **(ii)** We empirically demonstrate that this approach can perform on par with amortised models for generating long trajectories, and in particular that, unlike amortised models, using higher Markov order improves its performance. **(iii)** We show that this model can jointly tackle data assimilation, as well as forecasting, and in most settings outperforms the strategy introduced in Rozet & Louppe (2023a) which consists of sampling the full sequence at once. In summary, while this work does not introduce a novel methodology per se, we combine existing techniques to develop a PDE simulation method that is principled, accurate, and flexible. In addition to that, we perform a rigorous comparison

---

*Equal contribution

between amortisation and guidance in the context of PDE modelling, especially on the forecasting and data assimilation tasks, that no prior work has considered.

## 2 BACKGROUND

**Continuous-time diffusion models.** In this section we briefly recall the main concepts underlying continuous diffusion models, and refer readers to Song et al. (2021) for a thorough presentation. We consider a forward noising process $(x(t))_{t \geq 0}$ associated with the following stochastic differential equation (SDE),

$$\mathrm{d}x(t) = -\tfrac{1}{2}x(t)\mathrm{d}t + g_t \mathrm{d}w(t), \ x(0) \sim p_0, \tag{1}$$

with $w(t)$ an isotropic Wiener process, $p_0$ the data distribution, and $g_t$ a diffusion coefficient. The process defined by equation 1 is the well-known Ornstein–Uhlenbeck process, which geometrically converges to $\mathrm{N}(0, \mathrm{I})$. Importantly, under mild assumptions on $p_0$, the time-reversal process $(\overleftarrow{x}(t))_{t \geq 0}$ also satisfies an SDE (Cattiaux et al., 2022; Haussmann & Pardoux, 1986) which is given by

$$\mathrm{d}\overleftarrow{x}(t) = \left\{-\tfrac{1}{2}\overleftarrow{x}(t) - g_t^2 \nabla \log p_t(\overleftarrow{x}(t))\right\}\mathrm{d}t + g_t \mathrm{d}w(t),$$

where $p_t$ denotes the density of $x(t)$. In practice, the score $\nabla \log p_t$ is unavailable, and is approximated by a neural network $\mathbf{s}_\theta(t, \cdot) \approx \nabla \log p_t$, referred to as the *score network*. The parameters $\theta$ are learnt by minimising the denoising score matching (DSM) loss (Hyvärinen & Dayan, 2005; Vincent, 2011; Song et al., 2021)

$$\mathcal{L}(\theta) = \mathbb{E}[\|\mathbf{s}_\theta(t, x(t)) - \nabla \log p_t(x(t)|x(0))\|^2].$$

**Conditioning with diffusion models.** The methodology introduced above allows for (approximately) sampling from $x(0) \sim p_0$. Sampling from the conditional $p(x(0)|y)$, where $y$ represents some observations with likelihood $p(y|\mathrm{A}(x))$ and A a linear *measurement* operator, is more challenging. While by Bayes' rule we can rewrite it as $p(x(0)|y) = p(y|x(0))p(x(0))/p(y)$, this is yet not available in closed form.

Below we present two methods of simulating the *conditional* denoising process in Eq. (2) directly.

$$\mathrm{d}\overleftarrow{x}(t) = \left\{-\tfrac{1}{2}\overleftarrow{x}(t) - g_t^2 \nabla \log p_t(\overleftarrow{x}(t)|y)\right\}\mathrm{d}t + g_t \mathrm{d}w(t). \tag{2}$$

**Amortising over observations.** The simplest approach is to directly learn the conditional score $\log p_t(\overleftarrow{x}(t)|y)$ (Song et al., 2021; Ho et al., 2020). This can be achieved by additionally feeding the observations $y$ to the score network $\mathbf{s}_\theta$, whose parameters are then learnt by minimising the following DSM loss $\mathcal{L}(\theta) = \mathbb{E}[\|\mathbf{s}_\theta(t, x(t), y) - \nabla \log p_t(x(t)|x(0))\|^2]$, with $(x(0), y) \sim p_{0 \times y}$.

**Reconstruction guidance.** Alternatively, one can leverage Bayes rule, which allows to express the conditional score w.r.t. $x(t)$ as the sum of the following two gradients

$$\nabla \log p(x(t)|y) = \nabla \log p(x(t)) + \nabla \log p(y|x(t)) \approx s_\theta(t, x(t)) + \nabla \log q_\theta(y|x(t)). \tag{3}$$

The conditioning comes into play via the $\nabla \log p(y|x(t))$ term which is referred to as *reconstruction guidance* (Ho et al., 2022b; Chung et al., 2022; 2023; Meng & Kabashima, 2023; Wu et al., 2022; Song et al., 2022; 2023). Assuming a Gaussian likelihood $p(y|x(0)) = \mathrm{N}(y|\mathrm{A}x(0), \sigma_y^2\mathrm{I})$, we have

$$p(y|x(t)) = \int p(y|x(0))p(x(0)|x(t)) \, \mathrm{d}x(0) \approx \mathrm{N}\left(y|\mathrm{A}\hat{x}(x(t)), (\sigma_y^2 + r_t^2)\mathrm{I}\right) \triangleq q_\theta(y|x(t)) \tag{4}$$

where $p(x(0)|x(t))$ is not available in closed form and is approximated by a Gaussian $q(x(0)|x(t)) \triangleq \mathrm{N}(x(0)|\hat{x}(x(t)), r_t^2\mathrm{I}) \approx p(x(0)|x(t))$ with mean given by Tweedie's formula (Efron, 2011) and variance $\mathrm{Var}[x(0)|x(t)] \approx r_t^2$ being chosen as a monotonically increasing function (Rozet & Louppe, 2023a; Finzi et al., 2023; Song et al., 2022; Pokle et al., 2023).

## 3 SCORE-BASED FORECASTING AND DATA ASSIMILATION

**Problem setting** We denote by $\mathcal{P} : \mathcal{U} \to \mathcal{U}$ a differential operator taking functions $u : \mathbb{R}_+ \times \mathcal{Z} \to \mathcal{X} \in \mathcal{U}$ as input. Along with some initial $u(0, \cdot) = u_0$ and boundary $u(\tau, \partial \mathcal{Z}) = f$ conditions, these

define a partial differential equation (PDE) $\frac{\partial u}{\partial \tau} = \mathcal{P}(u; \alpha) = \mathcal{P}\left(\frac{\partial u}{\partial z}, \frac{\partial^2 u}{\partial z^2}, \ldots; \alpha\right)$, with coefficients $\alpha$. We assume to have access to many accurate numerical solutions from conventional solvers, which are discretised both in space and time, and denoted by $x_{1:L} = (x_1, \ldots, x_L) \in \mathbb{R}^{D \times L} \sim p_0$ with $D$ and $L$ being the size of the discretised space and time domains, respectively. The overall goal is to train a machine learning model on these solved trajectories and use it to solve similar PDEs with different initial condition $u_0$, boundary condition $f$ or parameter $\alpha$, as fast as possible while staying close to the true solution. We are generally interested in the problem of generating realistic PDE trajectories given observations, i.e. sampling from conditionals $p(x_{1:L}|y)$. Specifically, we focus on two tasks: (a) *forecasting*: predicting future states $x_{i:i+H}$ given some past states $x_{1:i-1}$, i.e. sampling from $p(x_{i:i+H}|x_{1:i-1})$, with $H$ the forecast *horizon*; (b) *data assimilation*: inferring the ground state of a dynamical system given some sparse observed states $x_o$, i.e. sampling from $p(x_{1:L}|x_o)$.

**Markovian dynamics.** Learning a diffusion model over the full joint distribution $p(x_{1:L}(t))$ can become prohibitively expensive for long trajectories, as it requires a score network taking the full sequence as input—$\mathbf{s}_\theta(t, x_{1:L}(t)) \approx \nabla_{x_{1:L}(t)} \log p(x_{1:L}(t))$, with memory footprint scaling linearly with the length $L$ of the sequence. One approach to alleviate this, as suggested by Rozet & Louppe (2023a), is to assume a Markovian structure of order $k$ such that the joint over noised data trajectories can be factorised into a series of conditionals $p(x_{1:L}) = p(x_1)p(x_2|x_1)\ldots p(x_{k+1}|x_{1:k}) \prod_{i=k+2}^{L} p(x_i|x_{i-k:i-1})$. Here we are omitting the time dependency $x_{1:L} = x_{1:L}(t)$ for clarity's sake. The score w.r.t. $x_i$ can be written as

$$\nabla_{x_i} \log p(x_{1:L}) = \nabla_{x_i} \log p(x_i|x_{i-k:i-1}) + \sum_{j=i+1}^{i+k} \nabla_{x_i} \log p(x_j|x_{j-k:j-1}) = \nabla_{x_i} \log p(x_{i-k:i+k})$$

with $k+1 \leq i \leq L-k-1$, whilst formulas for $i \leq k$ and $i \geq L-k$ can be found in App. B.1. Consequently, instead of learning the entire joint score at once $\nabla_{x_{1:L}(t)} \log p(x_{1:L}(t))$, we only need to learn the *local* scores $\mathbf{s}_\theta(t, x_{i-k:i+k}(t)) \approx \nabla_{x_{i-k:i+k}(t)} \log p_t(x_{i-k:i+k}(t))$ trained on short sequences of size $2k+1$. When necessary, the full score $\mathbf{s}_\theta(t, x_{1:L}(t)) \approx \nabla \log p_t(x_{1:L}(t))$ can then be reconstructed from these local scores—see App. B.1 and Rozet & Louppe (2023a, Alg 2.).

**Sampling.** We can generate samples $x_{1:L} \sim p(x_{1:L})$ by solving the time-reversal SDE over the sequence. Optionally, to avoid errors accumulating along the denoising discretisation, each *predictor* denoising step can be followed by a small number of Langevin Monte Carlo (*corrector*) steps.

## 3.1 Conditional sampling with guidance.

We now describe how to tackle the forecasting and data assimilation tasks by sampling from conditionals $p(x_{1:L}|x_o)$—instead of the prior $p(x_{1:L})$—with the trained local score $\mathbf{s}_\theta$ described in Sec. 3. The conditioning information $y = x_o \in \mathbb{R}^O$ in this case is a (possibly noisy) measurement of a subset of variables in the space-time domain, i.e. $x_o = \mathcal{A}(x) = A \text{vec}(x) + \eta$ with $\text{vec}(x) \in \mathbb{R}^N$ the vectorised trajectory, $\eta \sim \mathrm{N}(0, \sigma_y^2 I)$ and a masking matrix $A = \{0,1\}^{O \times N}$. Plugging this in (4) and computing the score we get the following reconstruction guidance term $\nabla \log p(x_o|x(t)) \approx \frac{1}{r_t^2 + \sigma_y^2}(y - A\hat{x}(x(t)))^\top A \frac{\partial \hat{x}(x(t))}{\partial x(t)}$. Summing up this guidance with the trained score over the prior as in (3), we get an approximation of the conditional score

$$\nabla_{x_{1:L}(t)} \log p(x_{1:L}(t)|x_0) = \nabla_{x_{1:L}(t)} \log p(x_{1:L}(t)) + \nabla_{x_{1:L}(t)} \log p(x_0|x_{1:L}(t))$$

$$\approx \mathbf{s}_\theta(t, x_{1:L}(t)) + \frac{1}{r_t^2 + \sigma_y^2}(y - A\hat{x}(x_{1:L}(t)))^\top A \frac{\partial \hat{x}(x_{1:L}(t))}{\partial x_{1:L}(t)}$$

and can thus generate conditional trajectories by simulating the conditional denoising process Eq. (2).

**All-at-once (AAO).** We can generate entire trajectories $x_{1:L}$ in one go as proposed by Rozet & Louppe (2023a). Assuming we observe some values $x_o \sim \mathrm{N}(\cdot|Ax_{1:L}, \sigma_y^2 I)$, the conditional score $\nabla \log p(x_{1:L}(t)|x_o)$ is obtained by summing the full sequence score $\mathbf{s}_\theta(t, x_{1:L}(t))$ built by combining local scores as developed in Sec. 3, with the guidance term $\nabla \log p(x_o|x_{1:L}(t))$ estimated as above.

**Autoregressive (AR).** Let's first focus on forecasting, i.e. sampling $x_{C+1:L}$ given $x_{1:C}$ initial states. An alternative approach to AAO is to factor the forecasting prediction problem as $p(x_{1:L}|x_{1:C}) = \prod_i p(x_i|x_{i-C:i-1})$ with $C \leq k$. As suggested in Ho et al. (2022b), it then follows that we can sample

the full trajectory by iteratively generating $H$ states conditioned on the previous $C$ ones, such that $H + C = 2k + 1$ with $k$ being the Markov order. We refer to this approach as GUARD for Guided AutoRegressive diffusion model. Data assimilation is similarly tackled by further conditioning on additional observations appearing within the predictive horizon $H$ at each AR step of the rollout.

**Modelling capacity.** There are two main reasons why the AR sampling strategy outperforms the AAO approach. First, for the same score network with an input window size of $2k + 1$, the AAO model has a Markov order $k$ whilst the AR model has an effective order of $2k$. Indeed, as shown above, a process of Markov order $k$ yields a score for which each component depends on $2k + 1$ inputs. Yet parameterising a score network taking $2k + 1$ inputs, means that the AR model would condition on the previous $2k$ states at each autoregressive step. This twice greater Markov order implies that the AR approach is able to model a strictly larger class of data processes than the AAO sampling— see App. B.1. Second, in AAO the denoising process must ensure coherence between the start and end of the sequence. This requires information to be propagated between the two ends of the sequence. However, at each AAO denoising step, the score network has a limited receptive field of $2k + 1$ which limits information propagation. In contrast, in AR sampling, there is no such limitation due to the nature of the autoregressive scheme. Scalability of the two methods is discussed in App. F.

## 3.2 AMORTISED MODEL.

One can directly learn an approximation to the *conditional* score. Practically, conditioning on the previous states $x_{i-C:i}$ is achieved by feeding them to the score network along with the input as separate channels (Voleti et al., 2022).

## 4 EXPERIMENTS

We parameterise the score network $s_\theta$ with a modern U-net architecture (Ronneberger et al., 2015; Gupta & Brandstetter, 2023) with residual connections and layer normalization. For the full details of our experimental setup we will refer to App. D, where more results can be found.

**Data.** In this experimental investigation, we look into the Burgers' and the Kuramoto-Sivashinsky (KS) equations since their dynamics are interesting yet challenging, and have been investigated in prior work (e.g. Zhang et al., 2019; Du & Zaki, 2021). The Kuramoto-Sivashinsky is a fourth-order nonlinear one-dimensional PDE describing flame fronts, whose position is governed by $\frac{\partial u}{\partial \tau} + u \frac{\partial u}{\partial z} + \frac{\partial^2 u}{\partial z^2} + \nu \frac{\partial^4 u}{\partial z^4} = 0$, where we used $\nu = 1$. We refer to App. C and App. D for more details on the data generating processes and results on the Burgers' equation.

**Evaluation metrics.** We use two sets of metrics to measure the accuracy of samples: (a) per time step—with the mean squared error $\text{MSE}_{1:L} = \mathbb{E}[(x_{1:L} - \hat{x}_{1:L})^2]$ and Pearson correlation $\rho_{1:L} = \frac{\text{Cov}(x_{1:L}, \hat{x}_{1:L})}{\text{Var}(x_{1:L})\text{Var}(\hat{x}_{1:L})}$ between model samples $\hat{x}_{1:L} \sim p_\theta$ and ground truth trajectories $x_{1:L} \sim p_0$, and (b) for the whole trajectory—via $\text{RSMD} = \sqrt{\frac{1}{L} \sum_{l=1}^{L} \text{MSE}_l}$ and *high correlation time* $t_{\max} = l_{\max} \times \Delta t$ where $l_{\max} = \arg\max_{l \in 1:L} \{\rho_l > 0.8\}$.

**Guided data assimilation.** In this section, we compare the two guided diffusion sampling regimes, AAO and AR—introduced in Sec. 3— with an amortized model on the data assimilation task. For each time step, we randomly sample a certain percentage of observed variables, i.e. these are sparse in space. We assume that we have some initial states that are fully observed. Our joint diffusion model is trained with a window of size 9 and in the AR case it generates 3 states at a time conditioning on the 6 previous ones. We vary the *sparsity* of observations, from almost fully observed to exclusively conditioning

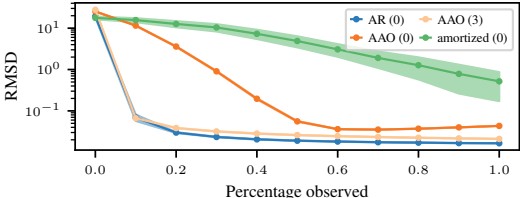

Figure 1: RMSD ($\downarrow$) computed on the KS dataset when varying the percentage of values observed. The value in bracket indicates the number of correction steps. Means and confidence intervals are estimated over 50 samples.

on the initial states, i.e. plain forecasting, and evaluate the different sampling regimes in terms of RMSD. We observe in Fig. 1 that apart from very dense observation regimes, the AR sampling

strategy outperforms AAO. We believe this is due to the effective Markov order being $k$ in AAO vs $2k$ for AR when using a local score network with a window of size $2k + 1$, as previously discussed. Correction steps are crucial for the AAO sampling strategy to work well, making it computationally intensive. For more details, see App. E. The conditional score, on the other hand, is significantly underperforming on this task.

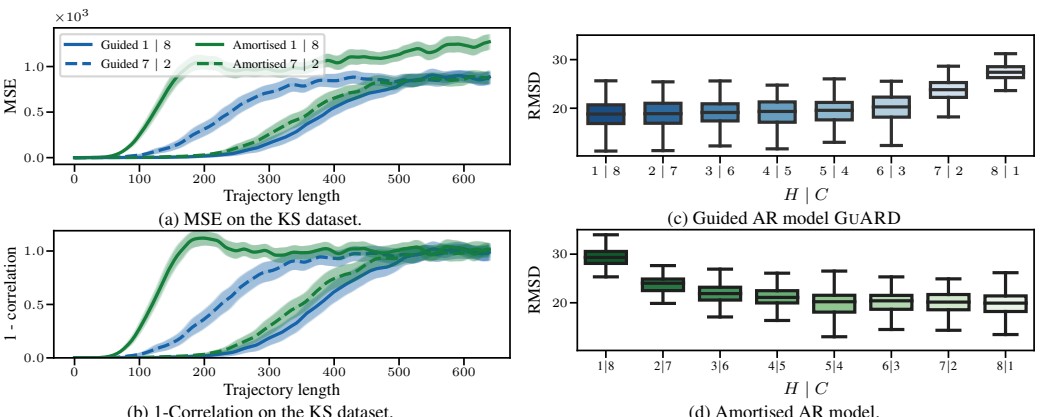

(a) MSE on the KS dataset.

(b) 1-Correlation on the KS dataset.

(c) Guided AR model GUARD

(d) Amortised AR model.

Figure 2: *(Left)* Evolution of MSE and correlation for the best amortised (7 | 2) and guided (1 | 8) models for a fixed window of size 9. *(Right)* RMSD ($\downarrow$) on the KS test dataset for different $H \mid C$ settings, with $H$ the number of states generated at each AR step, and $C$ the number of frames we condition upon.

**Autoregressive forecasting.**   We investigate the ability of trained diffusion models to sample long rollouts by starting from some fully observed noiseless initial states. We compare an *amortised* model trained on conditional states, and GUARD, a *guided* model trained on joint states. Both are trained over short sequences and sampled autoregressively. In this setting we are not reporting results for GUARD sampled all-at-once because it fails in generating long-rollouts even if several corrections steps are employed, see App. F.1. The models are trained on the KS equation dataset on short trajectories of length 140 and then evaluated on test trajectories of length 640. We use a fixed window size of 9 for both models. In Fig. 2, we observe that the best guided model (1 | 8, where 1 indicates the number of states generated and 8 the number of frames we condition on at each step) outperforms or performs on par with the best amortised model (7 | 2). Moreover, the guided model is not as sensitive to the conditioning length $C$ as the amortised one, as also outlined in App. H.2.3. We observe from Fig. 2 that the guided model is able to leverage longer past history to increase accuracy whilst the amortised model's performance quickly degrades as $C$ increases. In addition to that, the guided model is more flexible since one can choose $H \mid C$ at *sampling* time without the need to train a separate model as in the amortised case. More discussion and results are presented in App. D.

## 5   DISCUSSION

In this work, we have explored different ways to train and sample diffusion models for forecasting and data-assimilation tasks. Our proposed approach is a novel combination of existing techniques, such as learning a Markov score network (Rozet & Louppe, 2023b), amortising the score network (Song et al., 2021) using reconstruction guidance (Pokle et al., 2023) and autoregressive rollouts (Ho et al., 2022b). The paper serves as a proof that the combination of these techniques succeeds in modelling long-range rollouts of PDEs. In particular, we have empirically demonstrated the long-range rollout ability of diffusion models trained on short trajectory segments, and autoregressively sampled.

**Limitations and future work**   Although we have explored different solvers and a number of discretisation steps, the autoregressive sampling strategy requires simulating a denoising process at each step, which is computationally costly. We believe this could be further improved by taking advantage of the advancements in diffusion model research. Additionally, the guided sampling strategy requires tuning the guidance strength, yet we suspect there exist some simple heuristics that are able to decrease the sensitivity to this hyperparameter. Finally, in this work we only explored linear observation operators A, yet reconstruction guidance can also be employed with non-linear observation processes. We will investigate this in future work.

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

# SUPPLEMENTARY TO:

# GUIDED AUTOREGRESSIVE DIFFUSION MODELS WITH APPLICATIONS TO PDE SIMULATION

## A  ORGANISATION OF THE SUPPLEMENTARY

In this supplementary, we first motivate in App. B the choice made regarding the (local) score network given the Markov assumption on the sequences. We then describe the data generating process in App. C for the datasets used in the experiments in Sec. 4. In App. D, we present additional experiments that, because of the space limit, did not fit in the main paper. After, in App. E, we assess the ability of the all-at-once sampling strategy—which relies on reconstructing the full score from the local ones—to perform accurate forecast, and in particular show the crucial role of correction steps in the discretisation. Then in App. F, we show that the same learnt local score can give rise to two different sampling strategies—all-at-once and autoregressive—and empirically assess their performance for forecasting and data assimilation. In App. G, we look at how the guided autoregressive sampling approach behaves when trading off the prediction horizon with the number of states being conditioned over. Then in App. H, we compare the amortised model with the guided approach in the context of forecasting, and investigate how the window size and the combination between prediction horizon $H$ and number of conditioned frames $C$ affect the performance of each model. In App. I, we empirically explore different guidance schedules and their impact on forecast accuracy. Finally, in App. J we examine how sensitive the amortised and guided models are to noise in the initial observed states.

## B  MARKOV SCORE

In this section we look at the structure of the score induced by a Markov time series. In particular we lay out different ways to parameterise and train this score with a neural network. In what follows we assume an AR-1 Markov model for the sake of clarity, but the reasoning developed trivially generalises to AR-$k$ Markov models. Assuming an AR-1 Markov model, we have that the joint density factorise as

$$p(x_{1:L}) = p(x_1) \prod_{t=2}^{T} p(x_i|x_{i-1}). \tag{5}$$

### B.1  WINDOW $2k+1$:

Looking at the score of equation 5 for intermediary states we have

$$\nabla_{x_i} \log p(x_{1:L}) = \nabla_{x_i} \log p(x_{i+1}|x_i) + \nabla_{x_i} \log p(x_i|x_{i-1}) \tag{6}$$
$$= \nabla_{x_i} \log p(x_{i+1}, x_i|x_{i-1})$$
$$= \nabla_{x_i} \log p(x_{i+1}, x_i|x_{i-1}) + \underbrace{\nabla_{x_i} \log p(x_{i-1})}_{0}$$
$$= \nabla_{x_i} \log p(x_{i+1}, x_i, x_{i-1})$$
$$= \nabla_{x_i} \log p(x_{i-1}, x_i, x_{i+1}). \tag{7}$$

Similarly, for the first state we get

$$\nabla_{x_1} \log p(x_{1:L}) = \nabla_{x_1} \log p(x_1) + \nabla_{x_1} \log p(x_2|x_1)$$
$$= \nabla_{x_1} \log p(x_1, x_2)$$
$$= \nabla_{x_1} \log p(x_1, x_2) + \underbrace{\nabla_{x_1} \log p(x_3|x_2)}_{0}$$
$$= \nabla_{x_1} \log p(x_1, x_2) + \nabla_{x_1} \log \underbrace{p(x_3|x_2, x_1)}_{\text{since } = p(x_3|x_2)}$$
$$= \nabla_{x_1} \log p(x_1, x_2, x_3), \tag{8}$$

and for the last state

$$
\begin{aligned}
\nabla_{x_L} \log p(x_{1:L}) &= \nabla_{x_L} \log p(x_L|x_{L-1}) & (9) \\
&= \nabla_{x_L} \log \underbrace{p(x_L|x_{L-1}, x_{L-2})}_{\text{since } = p(x_L|x_{L-1})} \\
&= \nabla_{x_L} \log p(x_L|x_{L-1}, x_{L-2}) + \underbrace{\nabla_{x_L} \log p(x_{L-1}, x_{L-2})}_{0} \\
&= \nabla_{x_L} \log p(x_L, x_{L-1}, x_{L-2}) \\
&= \nabla_{x_L} \log p(x_{L-2}, x_{L-1}, x_L). & (10)
\end{aligned}
$$

**Noised sequence**  Denoising diffusion requires learning the score of *noised* states. The sequence $x_{1:L}$ is noised with the process defined by the forward equation introduced in Sec. 2 for an amount of time $t$: $x_{1:L}(t)|x_{1:L} \sim p_{t|0}$, where $p_{t|0}$ is the noising kernel. With $x_{1:L}$ an AR-$k$ Markov model there is no guarantee that $x_{1:L}(t)$ is still $k$-Markov. Yet, we can assume that $x_{1:L}(t)$ is an AR-$k'$ Markov model where $k' > k$, as in Rozet & Louppe (2023a). Although it may seem in contradiction with the previous statement, in what follows we focus on the AR-1 assumption since derivations and the general reasoning is much easier to follow, yet reiterate that this generalises to the order $k$.

**Local score**  Eqs. (7), (8) and (10) suggest training a score network to fit $\mathbf{s}_\theta(t, x_{i-1}(t), x_i(t), x_{i+1}(t)) \approx \nabla \log p_t(x_{i-1}(t), x_i(t), x_{i+1}(t))$ by sampling tuples $x_{i-1}, x_i, x_{i+1} \sim p_0$ and noising them.

**Full score**  The first, intermediary and last elements of the score are then respectively given by

- $\nabla_{x_1(t)} \log p_t(x_{1:L}(t)) \approx \mathbf{s}_\theta(t, x_1(t), x_2(t), x_3(t))[1]$
- $\nabla_{x_i(t)} \log p_t(x_{1:L}(t)) \approx \mathbf{s}_\theta(t, x_{i-1}(t), x_i(t), x_{i+1}(t))[2]$
- $\nabla_{x_L(t)} \log p_t(x_{1:L}(t)) \approx \mathbf{s}_\theta(t, x_{L-2}(t), x_{L-1}(t), x_L(t))[3]$

**Joint sampling**  The full score over the sequence $x_{1:L}$ can be reconstructed via the trained local score network taking segments of length 3 as input. This allows for sampling from the joint. All of the above generalise to AR-$k$ models, where the local score network takes a segment of length $2k+1$ as input.

**Guided AR**  Alternatively for forecasting, having learnt a local score network $\mathbf{s}_\theta(t, x_{i-1}(t), x_i(t), x_{i+1}(t))$, i.e. modelling joints $p(x_{i-1}, x_i, x_{i+1})$, we can condition on $x_{i-1}, x_i$ to sample $x_{i+1}$ leveraging reconstruction guidance—as described in Sec. 3. Effectively, this induces an AR-2—instead of AR-1— Markov model, or in general, an AR-$2k$—instead of $k$—Markov model. As such, this score parameterisation takes more input than strictly necessary to satisfy the $k$ Markov assumption.

**Amortised AR**  Obviously we note that, looking at equation 6 and equation 9, one could learn a conditional score network amortised on the previous value such that $\mathbf{s}_\theta(t, x_i(t)|x_{i-1}(0)) \approx \nabla_{x_i(t)} \log p(x_i(t)|x_{i-1}(0))$. This unsurprisingly allows forecasting with autoregressive rollout as in Sec. 3.2. Additionally learning a model over $p(x_1)$ and summing pairwise the scores as in equation 6 would also allow for joint sampling.

### B.2  WINDOW $k+1$:

In App. B.1, we started with an AR-$k$ model, but ended up with learning a local score taking $2k+1$ inputs. Below we derive an alternative score parameterisation. Still assuming an AR-1 model equation 5, we have that the score for intermediary states can be expressed as

$$
\begin{aligned}
\nabla_{x_i} \log p(x_{1:L}) &= \nabla_{x_i} \log p(x_{i+1}|x_i) + \nabla_{x_i} \log p(x_i|x_{i-1}) \\
&= \underbrace{\nabla_{x_i} \log p(x_i|x_{i+1}) - \nabla_{x_i} \log p(x_i)}_{\text{via Bayes' rule}} + \nabla_{x_i} \log p(x_i|x_{i-1}) + \underbrace{\nabla_{x_i} \log p(x_{i-1})}_{0} \\
&= \nabla_{x_i} \log p(x_i|x_{i+1}) + \underbrace{\nabla_{x_i} \log p(x_{i+1})}_{0} - \nabla_{x_i} \log p(x_i) + \nabla_{x_i} \log p(x_i, x_{i-1}) \\
&= \nabla_{x_i} \log p(x_i, x_{i+1}) - \nabla_{x_i} \log p(x_i) + \nabla_{x_i} \log p(x_i, x_{i-1}). & (11)
\end{aligned}
$$

**Local score**  From equation 11, we notice that we could learn a pairwise score network such that $\tilde{\mathbf{s}}_\theta(t, x_{i-1}(t), x_i(t)) \approx \nabla_{x_i(t)} \log p(x_{i-1}(t), x_i(t))$, and $\tilde{\mathbf{s}}_\theta(t, x_i(t)) \approx \nabla_{x_i(t)} \log p(x_i(t))$, trained by sampling tuples $x_{i-1}, x_i, x_{i+1} \sim p_0$ and noising them. Note that this requires 3 calls to the local score network instead of 1 as in App. B.1.

**Full score**  The first, intermediary and last elements of the score are then respectively given by

$$\nabla_{x_1(t)} \log p(x_{1:L}(t)) = \nabla_{x_1(t)} \log p(x_1(t)) + \nabla_{x_1(t)} \log p(x_2(t)|x_1(t))$$
$$= \nabla_{x_1(t)} \log p(x_1(t), x_2(t))$$
$$\triangleq \tilde{\mathbf{s}}_\theta(t, x_1(t), x_2(t))[1].$$

$$\nabla_{x_i(t)} \log p(x_{1:L}(t)) = \mathbf{s}_\theta(t, x_{i-1}(t), x_i(t), x_{i+1}(t))$$
$$\triangleq \tilde{\mathbf{s}}_\theta(t, x_i(t), x_{i+1}(t))[1] + \tilde{\mathbf{s}}_\theta(t, x_{i-1}(t), x_i(t))[2] - \tilde{\mathbf{s}}_\theta(t, x_i(t)).$$

$$\nabla_{x_L(t)} \log p(x_{1:L}(t)) = \nabla_{x_L(t)} \log p(x_L(t)|x_{L-1}(t))$$
$$= \nabla_{x_L(t)} \log p(x_L(t), x_{L-1}(t))$$
$$= \tilde{\mathbf{s}}_\theta(t, x_{L-1}(t), x_L(t))[2].$$

**Joint sampling**  The full score over $x_{1:L}(t)$ could therefore be reconstructed from these, enabling joint sampling.

**AR**  Additionally, autoregressive forecasting could be achieved with guidance by iterating over pairs $(x_{i-1}, x_i)$, thus using a local score of window size $k + 1$, in contrast to App. B.1 which requires an inflated window of size $2k + 1$.

## C  DATA GENERATION

### C.1  BURGERS' EQUATION

Groundtruth trajectories for the Burgers' equation (Burgers, 1948)

$$\frac{\partial u}{\partial \tau} + u \frac{\partial u}{\partial z} = \nu \frac{\partial^2 u}{\partial z^2},$$

are obtained from the open-source dataset made available by Li et al. (2021), where they consider $(x, \tau) \in (0, 1) \times (0, 1]$. The dataset consists of 1200 trajectories of 101 timesteps, i.e. $\Delta\tau = 0.01$, and where the spatial discretisation used is 128. 800 trajectories were used as training set, 200 as validation set and the remaining 200 as test set. These trajectories were generated following the setup described in (Wang et al., 2021) using the code available in their public repository [1]. Initial conditions are sampled from a Gaussian random field defined as $N(0, 625^2(-\Delta + 5^2 I)^{-4})$ (although be aware that in the paper they state to be using $N(0, 25^2(-\Delta + 5^2 I)^{-4})$, which is different from what they are doing in the code) and then they integrate the Burgers' equation using spectral methods up $\tau = 1$ using a viscosity of 0.01. Specifically, they solve the equations using a spectral Fourier discretisation and a fourth-order stiff time-stepping scheme (ETDRK4) (Cox & Matthews, 2002) with step-size $10^{-4}$ using the MATLAB Chebfun package (Driscoll et al., 2014). Examples of training, validation, and test trajectories are shown in Fig. 3.

### C.2  1D KURAMOTO-SIVASHINSKY

We generate the groundtruth trajectories for the 1D Kuramoto-Sivashinsky equation

$$\frac{\partial u}{\partial \tau} + u \frac{\partial u}{\partial z} + \frac{\partial^2 u}{\partial z^2} + \nu \frac{\partial^4 u}{\partial z^4} = 0,$$

following the setup of Lippe et al. (2023), which is based on the setup defined by Brandstetter et al. (2022). In contrast to Lippe et al. (2023), we generate the data by keeping $\Delta x$ and $\Delta\tau$ fixed, i.e.

---

[1] https://github.com/PredictiveIntelligenceLab/Physics-informed-DeepONets/tree/main/Burger/Data

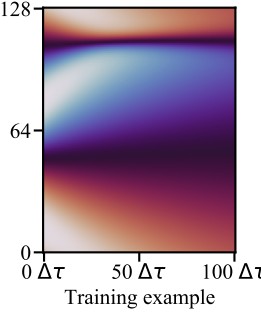 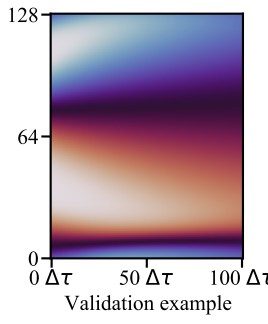 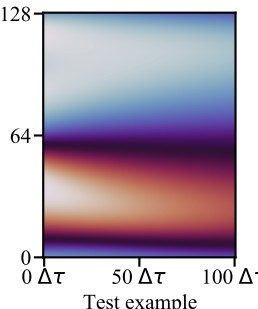

Figure 3: True test trajectories from solving the Burgers' equation following the setup of (Wang et al., 2021). In this case both training, validation, and test trajectories have the same length. We used $\Delta\tau = 0.01s$, so the trajectory contain states from time $\tau = 0s$ to $\tau = 1s$. The spatial dimension is discretized in 128 evenly distributed states

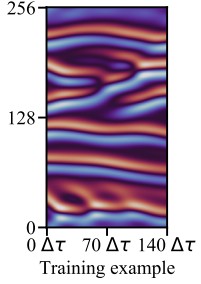 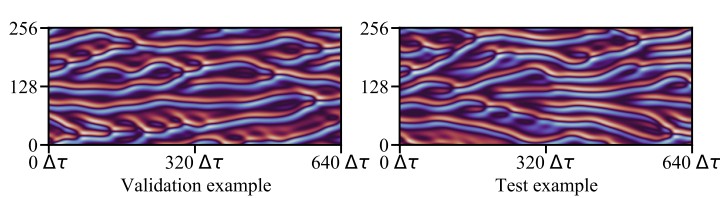

Figure 4: Examples from the Kuramoto-Sivashinsky dataset. Training trajectories are shorter than those used to evaluate the models. Training trajectories has 140 time steps generated every $\Delta\tau = 0.2s$, i.e. training trajectories contains examples from $0s$ to $28s$. Validation and test trajectories, instead, show the evolution of the equation for 640 steps, i.e. from $0s$ to $128s$. The spatial dimension is discretized in 256 evenly distributed states.

using a constant time and spatial discretisation[2]. The public repository of Brandstetter et al. (2022) can be used to generate our dataset by using the same arguments as the one defined in Lippe et al. (2023, App D1.).

Brandstetter et al. (2022) solves the KS-equation using the method of lines, with spatial derivatives computed using the pseudo-spectral method. Initial conditions are sampled from a distribution defined over truncated Fourier series, i.e. $u_0(x) = \sum_{k=1}^{K} A_k \sin(2\pi l_k x/L + \phi_k)$, where $A_k, l_k, \phi_{k_k}$ are random coefficients representing the amplitude of the different sin waves, the phase shift, and the space dependent frequency. The viscosity parameter is set to $\nu = 1$. Data is initially generated with `float64` precision and then converted to `float32` for training the different models. We generated 1024 trajectories of $140\Delta\tau$, where $\Delta\tau = 0.2s$, as training set, while the validation and test set both contain 128 trajectories of length $640\Delta\tau$. The spatial domain is discretised into 256 evenly spaced points. Trajectories used to train and evaluate the models can be seen in Fig. 4.

## D  ADDITIONAL EXPERIMENTS

In this section we present additional experiments performed on the Burgers' equation dataset and additional results that did not fit into the main text of the paper. As we have already mentioned, we parameterise the score network $\mathbf{s}_\theta$ with a modern U-net architecture (Ronneberger et al., 2015; Gupta & Brandstetter, 2023) with residual connections and layer normalisation. The U-net we used closely follows the architecture used in (Lippe et al., 2023). If not stated differently, we generate conditional samples by simulating the reverse diffusion ODE using the DPM solver (Lu et al., 2023) in 128 evenly spaced discretisation steps. As a final step, we return the posterior mean over the noise free data via

---

[2]This just requires changing lines 166-171 of the `generate_data.py` file in the public repository of Brandstetter et al. (2022).

Tweedie's formula. Unless specified otherwise, we set the guidance schedule to $r_t^2 = \gamma \sigma_t^2 / \mu_t^2$ (Finzi et al., 2023; Rozet & Louppe, 2023a), tune $\gamma$ via grid-search and set $\sigma_y = 0.01$.

## D.1 BURGERS' DATASET RESULTS

We now present the results obtained on the Burgers' equation dataset. The Burgers' equation is a second-order nonlinear PDE describing the motion of viscous fluid or gas with speed $u$ given by $\frac{\partial u}{\partial \tau} + u \frac{\partial u}{\partial z} = \nu \frac{\partial^2 u}{\partial z^2}$, where $\nu > 0$ is the viscosity. In our experiments we consider $\nu = 0.01$. As done in Sec. 4, we train a local score on contiguous segments of size 9 randomly sampled from training trajectories. We evaluate the model on test trajectories of length 101, which is the same length as the training set trajectories in the case of the Burgers' equation dataset. We assume we have access to the true noiseless initial observations for forecasting. Fig. 5 shows examples of sampled trajectories from both the guided and the amortised models. We can see that for the Burgers' equation both models are able to generate accurate forecasts.

By comparing one of the best guided models (1 | 8) with one of the best amortised models (7 | 2)[3], we can see that the former leads to better results in terms of MSE and correlation on the considered test trajectories. As in the KS dataset, we see that the guided model leads to better results when we condition on many previous states. On the other hand, the amortised model seems to perform better when the number of conditioned states $C$ is small. Similar to what we discovered in the KS dataset, we can see that the guided model is more robust to the number of conditioning states $C$ used at sampling time. Using one of worst performing $H \mid C$ guided configurations still leads to good performance (dashed blue line in Fig. 5 - Guided 7 | 2). In contrast, when we use one of the worst $h \mid C$ configurations for the amortised model, the metrics degrade quickly with trajectory length (solid green line in Fig. 5 - Amortised 1 | 8).

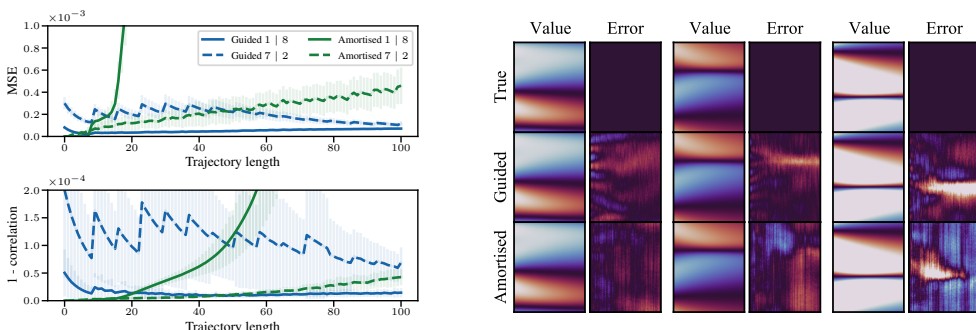

Figure 5: *(Left)* Evolution of MSE and correlation for the best amortised (7 | 2) and guided (1 | 8) models for a fixed window of size 9. *(Right)* Forecasting rollouts on the Burgers' equation dataset for guided and amortised autoregressive sampling strategies. The first 6 initial states are known.

## D.2 ADDITIONAL RESULTS ON THE KS EQUATION

**Example of a sampled trajectory.** In Fig. 6, we present an example test trajectory and the corresponding trajectories generated by the guided and the amortised models when conditioned on the first six states. We observe that both models generate similar trajectories up to 60 seconds. After that, the error accumulation makes the trajectories diverge from the true one.

**Observation noise and guidance ablation.** We further explore different variants of the amortised and guided AR models. Specifically, the reconstruction guidance term

$$\nabla_{x(t)} \log p(y|x(t)) \approx \frac{1}{\gamma r_t^2 + \sigma_y^2} \nabla_{x(t)} \|y - A\hat{x}(x(t))\|^2,$$

involves a choice of *guidance schedule* $r_t^2 \approx \text{Var}[x(0)|x(t)]$, and a *guidance scaling* $\gamma$ to be tuned. We explore several heuristics proposed in the literature:

---

[3]The guided and amortised models showed similar performance for several configurations (within error bars). This is why we state that we show the results for one of the best configuration for each corresponding model.

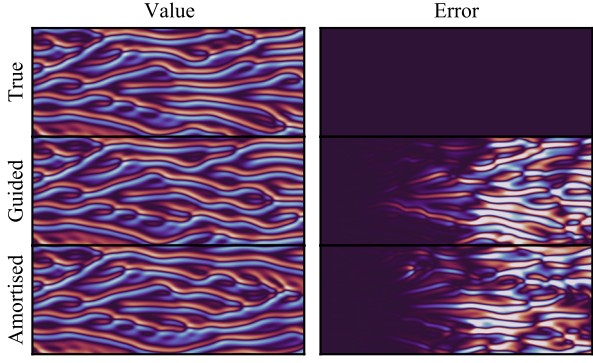

Figure 6: Forecasting rollouts for guided and amortised autoregressive sampling strategies on the KS dataset. The first 6 initial states are known.

- SDA: $r_t^2 = \frac{\sigma_t^2}{\mu_t^2}$, $\sigma_y = 0.01$ (Rozet & Louppe, 2023a; Finzi et al., 2023)

- ΠGDM: $r_t^2 = \frac{\sigma_t^2}{\mu_t^2 + \sigma_t^2}$, $\sigma_y = 0.01$ (Pokle et al., 2023; Song et al., 2022)

- DPS: $r_t^2 = \|y - A\hat{x}(x(t))\|$, $\sigma_y = 0$ (Chung et al., 2023)

- Video diffusion: $r_t^2 = 1/\mu_t$, $\sigma_y = 0$ (Ho et al., 2022b).

Although we do not know the exact variance, we know that $r_t^2 = \text{Var}[x(0)|x(t)] \xrightarrow[t\to 0]{} 0$, and $\text{Var}[x(0)|x(t)] \xrightarrow[t\to\infty]{} \text{Var}[x(0)] = \sigma_0^2$ and that the variance should be monotonically increasing with $t$. For SDA and ΠGDM, we have that $r_0 = 0$, whilst $r_0 = 1$ for Video diffusion and $r_t \xrightarrow[t\to 0]{} \infty$ for DPS (assuming the error goes to 0). Also, for SDA and Video diffusion, $r_t \xrightarrow[t\to\infty]{} \infty$ whilst $r_t \xrightarrow[t\to\infty]{} 1$ for ΠGDM. These properties make the ΠGDM and SDA guidance schedules reasonable heuristics to choose, and indeed from Fig. 7 we observe that these perform best.

For the amortised model, we also consider adding Gaussian noise $\eta \sim \text{N}(0, s^2)$ to the conditional frames during training, with a fixed noise $s = 0.01$. On Fig. 7 we see that noising the conditional information hurts the amortised model's performance, although one might think that it could improve robustness w.r.t. error accumulation. Also, App. J shows that the amortised model trained without noise is much more sensitive to noisy initial condition when sampling compared to GUARD and the amortised model trained with $s = 0.01$.

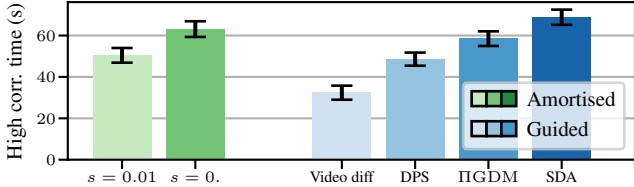

Figure 7: High correlation time on the Kuramoto-Sivashinsky equation. Means and standard errors are estimated over 128 sampled trajectories. The guided models using ΠGDM, Video diff, and DPS were trained on rescaled data, such that dynamic thresholding over the $x_0$ prediction could be employed to ensure stability; otherwise, the performance was very poor (see App. I).

## E    ALL-AT-ONCE JOINT SAMPLING FOR FORECASTING

One approach for generating long rollouts conditioned on some observations using score-based generative modelling is to directly train a joint posterior score $s_\theta(t, x_{1:L}(t)|y)$ to approximate $\nabla_{x_{1:L}(t)} \log p(x_{1:L}(t)|y)$. The posterior can be composed from a prior score and a likelihood term, just as in the main paper. However, this approach cannot be used for generating long rollouts (large

$L$), as the score network becomes impractically large. Moreover, depending on the dimensionality of each state, with longer rollouts the probability of running into memory issues increases significantly. To deal with this issue, in this section we develop the approach proposed by Rozet & Louppe (2023a), whereby the prior score $(s_\theta(t, x_{1:L}(t)))$ is approximated with a series of local scores $(s_\theta(t, x_{i-k:i+k}(t)))$, which are easier to learn. The size of these local scores is in general significantly smaller than the length of the generated trajectory and will be denoted as the window of the model $W$.

At sampling time, the entire trajectory is generated all-at-once, with the ability to condition on initial and/or intermediary states. We study two sampling strategies. The first one is based on the exponential integrator (EI) discretization scheme (Zhang & Chen, 2023), while the second one uses the DPM++ solver (Lu et al., 2023). For each strategy, the predictor steps can be followed by a few corrector steps (Langevin Monte Carlo) (Song et al., 2021; Mathieu et al., 2023). For the results below, we consider the case of conditional generation, where we generate trajectories of 101 states, conditioned on a noisy version of the initial 6 states (with standard deviation $\sigma_y = 0.01$). The results are computed for the Burgers' dataset, based on 30 test trajectories, each with different initial conditions. The sampling procedure uses 128 diffusion steps and a varying number of corrector steps.

**Influence of solver and time scheduling**  We show the results using two solvers: EI (Zhang & Chen, 2023) and DPM++ (Lu et al., 2023). For each, we study two time scheduling settings, which determine the spacing between the time steps during the sampling process

- Linear/uniform spacing: $t_i = t_N - (t_N - t_0)\frac{i}{N}$
- Quadratic: $t_i = (\frac{N-i}{N}t_0^{1/\kappa} + \frac{i}{N}t_N^{1/\kappa})^\kappa$

where $N$ indicates the number of diffusion steps, $t_N$ and $t_0$ are chosen to be 1 and $10^{-3}$ and $\kappa$ is chosen to be 2.

Fig. 8 shows the RMSD between the generated samples and the true trajectories, averaged across the trajectory length. It indicates that for AAO sampling with 0 corrections, there is little difference between the two time schedules, and that DPM++ slightly outperforms EI in terms of RMSD. However, the differences are not significant when taking into account the error bars. As the number of corrections is increased, the difference in performance between the two solvers gets negligible. However, it does seem that with a large number of corrections (25), the quadratic time schedule leads to lower RMSD. Nevertheless, given that, in general, only 3-5 corrector steps are used, we argue that the choice of solver and time schedule has a small impact on overall performance.

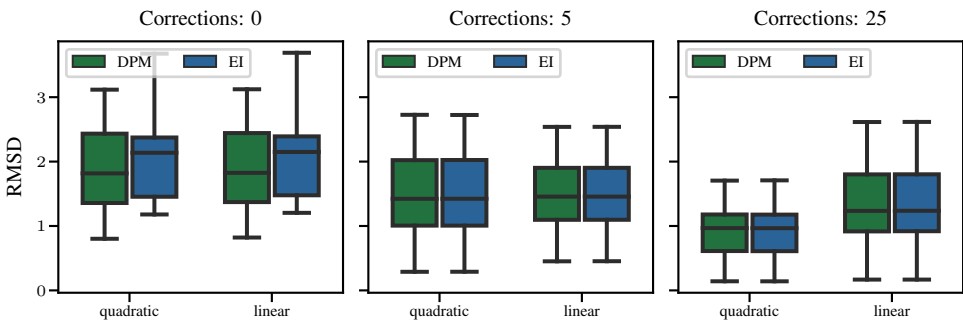

Figure 8: RMSD for Burgers' for the two solvers (DPM++ and EI), and the two time schedules (quadratic and linear) used. The results are shown for 0 (left), 5 (middle), and 25 (right) corrector steps. When using no/only a few corrector steps, the RMSD is not highly impacted by the choice of solver and time schedule. However, at 25 corrector steps, the quadratic time schedule seems to lead to lower RMSD.

**Influence of corrector steps and window size**  Figs. 9 and 10 illustrate the effect of the window size and number of corrector steps on the mean squared error (MSE) and the correlation coefficient (as compared to the ground truth samples). Based on the findings from above, here we used the DPM++ solver with quadratic time discretisation.

The corrector steps seem to be crucial to prevent the error accumulation, with performance generally improving with increasing corrections for both models (window 5 and 9). However, this comes at

a significantly higher computational cost, as each correction requires one extra function evaluation. Unsurprisingly, the bigger model (window 9) is better at capturing time dependencies between states, but it comes with increased memory requirements.

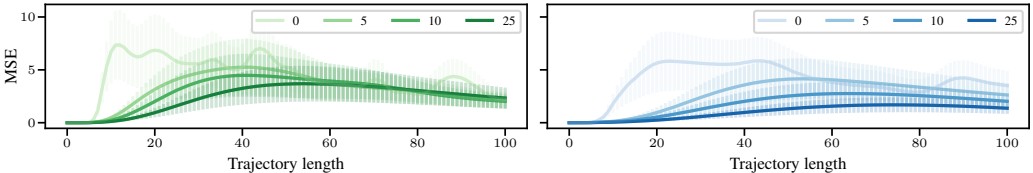

Figure 9: Evolution of the MSE for Burgers' along the trajectory length for a window of 5 (left) and 9 (right). The error bars indicate ± 3 standard errors. Each line corresponds to a different number of corrector steps. Unsurprisingly, the window 9 model performs better than the window 5 model (when using the same number of corrector steps). Performing corrector steps, alongside predictor steps, is crucial for preventing errors from accumulating. However, even with 25 corrector steps the performance is not very good.

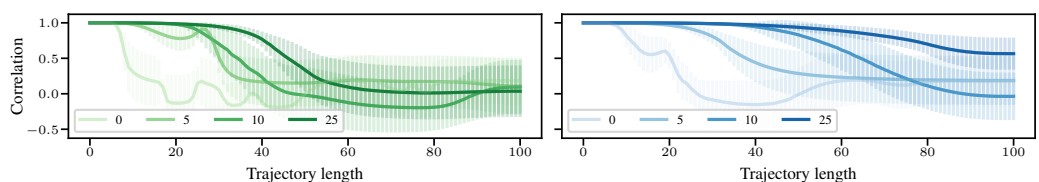

Figure 10: Evolution of the Pearson correlation coefficient for Burgers' along the trajectory length for a window of 5 (left) and 9 (right). The error bars indicate ± 3 standard errors. The findings are entirely consistent with the ones suggested by the MSE results.

Finally, the findings from above are also confirmed in Fig. 11, which illustrates that the predictive performance increases with the window size and the number of corrector steps. Both of these come at an increased computational cost; increasing the window size too much might lead to memory issues, while increasing the number of corrections has a negative impact on the sampling time.

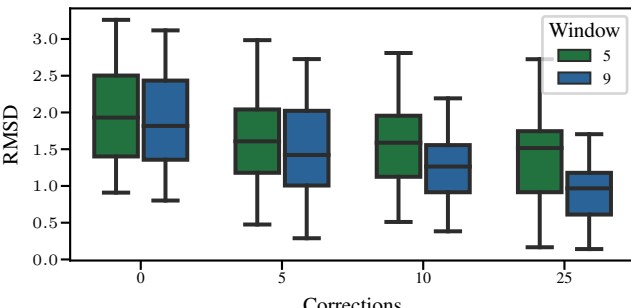

Figure 11: RMSD for Burgers' as a function of corrector steps for the two studied models (window 5 and window 9). Increasing both the window size and the number of corrections positively impacts the predictive performance.

## F    COMPARISON BETWEEN ALL-AT-ONCE AND AUTOREGRESSIVE SAMPLING

Once the joint model $s_\phi(t, x_{1:L}(t)|y) \approx \nabla_{x_{1:L}(t)} \log p(x_{1:L}(t)|y)$ is trained, there are multiple ways in which it can be queried

- All-at-once (AAO): sampling the entire trajectory in one go (potentially conditioned on some initial and/or intermediary states);
- Autoregressively (AR): sampling a few states at a time, conditioned on a few previously generated states (as well as potentially on some initial and/or intermediary states).

Let us denote the length of the generated trajectory with $L$, the number of predictor steps used during the sampling process with $p$, and the number of corrector steps with $c$. Moreover, in the case of the AR procedure, we have two extra hyperparameters: the predictive horizon $H$ (the number of states generated at each iteration), and the number of conditioned states $C$. Finally, let the window of the model be $W$. Each sampling procedure has its advantages and disadvantages.

1. Sampling time: the AAO procedure is faster than the AR one due to its non-recursive nature. We express the computational cost associated with each procedure in terms of the number of function evaluations (NFEs) (i.e. number of calls of the neural network) needed to generate the samples.

   - **AAO**: NFE = $(1 + c) \times p$
   - **AR**: NFE = $(1 + c) \times p \times (1 + \frac{L-W}{H})$

2. Memory cost: the AAO procedure is more memory-intensive than the AR one, as the entire generated trajectory length needs to fit in the memory when batching is not performed. If batching is employed, then the NFEs (and hence, sampling time) increases. In the limit that only one local score evaluation fits in memory, the AAO and AR procedures require the same NFEs.

3. Redundancy of generated states: in the AR procedure, at each iteration the model re-generates the states it conditioned upon, leading to computational redundancy.

4. Quality of generated samples: the AR procedure manages to capture long-term temporal dependencies between states better than the AAO procedure. One of the main reasons likely has to do with the fact that AR sampling induces an AR-$2k$ Markov model, as opposed to a AR-$k$ model in the AAO case (see App. B).

### F.1 GUIDED FORECASTING

This section aims to illustrate a fair comparison between the AAO and AR sampling methodologies using the Burgers' dataset in the context of forecasting. As mentioned above, the AR method comes at an increased computational cost at sampling time, but generates samples of higher quality and manages to maintain temporal coherency even during long rollouts. However, as illustrated in App. E, the quality of the AAO procedure can be improved by using multiple corrector steps. In particular, to obtain the same NFE for AAO and AR sampling (assuming the same number of predictor steps $p$ and no batching for AAO), we can use $c = \frac{L-W}{H}$ corrector steps.

**Window = 5** Fig. 12 shows the results for a trajectory with $L = 101$, using a model with $W = 5$. The results are expressed in terms of the RMSD, calculated based on 30 test trajectories. For AAO sampling $c = 48$, while for AR sampling $c = 0$. In the AR sampling, we generated 2 states at a time conditioned on the 3 previously generated states. In both cases we conditioned on a noisy version of the initial 3 true states (with standard deviation $\sigma_y = 0.01$).

For all experiments, we used the EI discretization scheme, with a quadratic time schedule. The AR

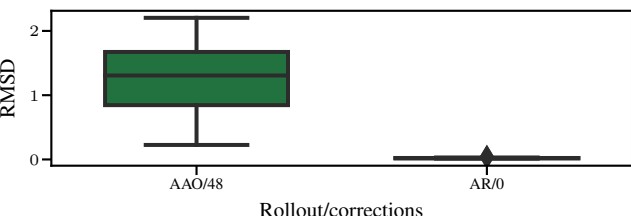

Figure 12: RMSD for Burgers' for AAO and AR generation with a window of 5. The number of corrections for AAO generation was chosen to match the computational budget of the AR sampling technique, yet the AR technique is clearly superior.

procedure is clearly superior to the AAO, even with 48 corrector steps. This can be easily observed by comparing the generated samples to the ground truth as shown in Fig. 13.

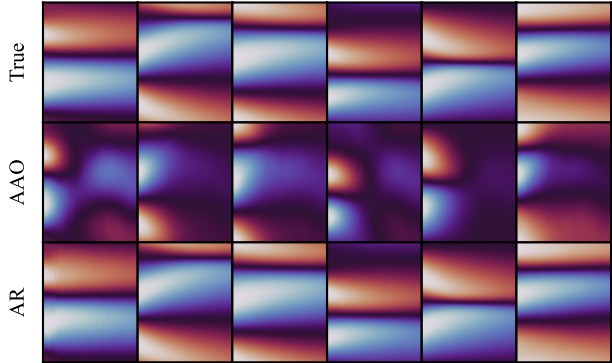

Figure 13: Comparison between the true samples (top), AAO samples (middle), and AR samples (bottom) for Burgers'. The AAO are only able to capture the initial states correctly, while the AR-generated samples maintain very good correlation with the ground truth even at long rollouts (101 time steps).

**Window = 9**   For a window of 9, we used the following number of corrector steps for each conditioning scenario:

- $H = 2, C = 7 : c = \frac{101-9}{2} = 46$
- $H = 3, C = 6 : c = \frac{101-9}{3} = 31$
- $H = 4, C = 5 : c = \frac{101-9}{4} = 23$
- $H = 5, C = 4 : c = \frac{101-9}{5} = 19$

Moreover, the number of states we initially condition upon is equal to $C$ (i.e., when comparing AAO with 46 corrections to AR with $H = 2, C = 7$ (2 | 7), we condition on the first 7 initial noised up states).

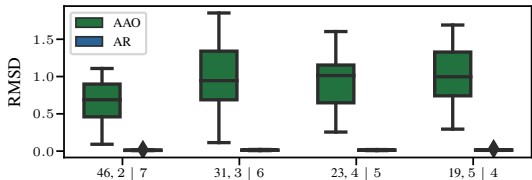

Figure 14: RMSD for Burgers' for AAO and AR generation with a window of 9 and with a varying computational budget. In the AAO case the computational budget is dictated by the number of corrector steps (indicated on the x-axis through the number before the comma). In the AR case, the budget is determined by the number of states generated at each iteration ($H$). This is also indicated on the x-axis in the format $H \mid C$ (e.g., 2 | 7 implies we generate two states at a time and condition on 7). AR is clearly superior to AAO for forecasting.

Once again, the AR-generated trajectories achieve significantly better metrics, which also reflects in the quality of the generated samples. Fig. 15 shows examples for AAO: 46/AR: 2 | 7.

## F.2   GUIDED DATA ASSIMILATION

Performing data assimilation using the AAO sampling procedure simply corresponds to conditioning at generation time on all observed variables regardless of the time-step $\tau$ they appear at using reconstruction guidance. This is possible because in AAO all the states of the trajectory are generated simultaneously. In the AR case, instead, we sample a full trajectory in multiple rollout steps. At each step we generate just $H + C$ states, where $C$ refers to the previous states we are conditioning on and $H$ to the number of new states we generate in one iteration. In the forecasting setting, we keep the $H$ newly generated states and discard the $C$ conditioning states. When performing data assimilation, we need to condition on both the previous $C$ states, as well as the observations appearing within the predictive horizon $H$ at each rollout step. If there are no observations in $H$, then we just perform

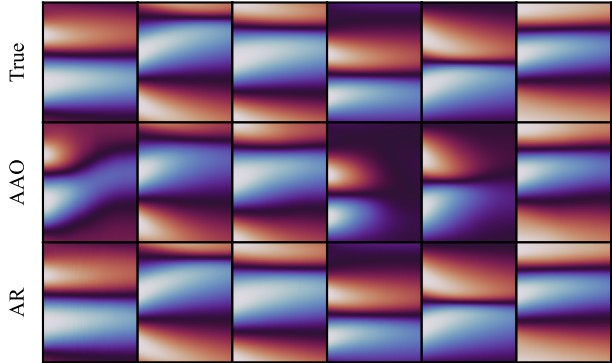

Figure 15: Comparison for Burgers' between the true samples (top), AAO samples with 46 corrections (middle), and AR 2 | 7 samples (bottom). With 46 corrector steps, some AAO trajectories manage to maintain good correlation with some of the easier test trajectories. However, the quality of the AR samples is clearly superior to the AAO samples, managing to model all test trajectories well.

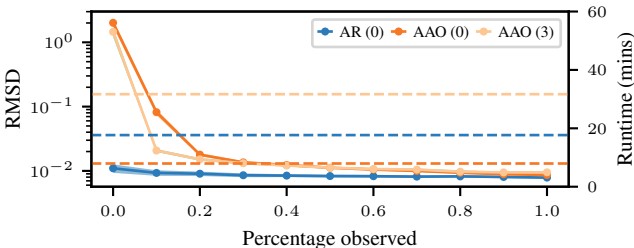

Figure 16: RMSD computed on the Burgers' dataset when varying the percentage of values observed at each physical timestep. The value between brackets indicates the number of correction steps. Means and confidence intervals are estimated over 50 samples. Dashed lines represent the runtime for each method computed on a Titan V with 12GB of memory.

a forecast step. If instead some observed variables are present in the predictive horizon, we have a likelihood for the conditioning states and one for the true observations within $H$. In this case, we also keep the resampled $C$ states to ensure more coherency between nearby states. In this section, since we follow the data assimilation set-up from (Rozet & Louppe, 2023b), we have observations at each time step and therefore we never perform any plain forecast rollout in the data assimilation experiments.

**Results on Burgers' equation dataset** In Sec. 4 we presented the results for the data assimilation task on the KS dataset. A similar behaviour can be also found when performing data assimilation on Burgers'. Burgers' equation, while still being non-linear, is less chaotic than the Kuramoto-Sivashinsky, resulting in an easier task for data assimilation when the setup is the same as the one presented in Sec. 4. This can be noticed in Fig. 16, where the main performance differences between the considered approaches are mostly in the very sparse observation setting (percentage observed of up to 20%). In addition to that, the use of correction steps is not as important as in the KS dataset.

**Runtime comparison between AAO and AR for DA** In Sec. 4, we presented results for the data-assimilation task on the KS equation dataset. In Fig. 17, we report again the results considering but mostly focusing on the two guided approaches, AAO and AR. We compare them also in terms of their runtime. While using corrections is beneficial and needed for AAO, this leads to an increased runtime than the AR procedure.

**Impact of the guidance scaling in the data assimilation task** The performance in the DA task seems to be slightly influenced by the value of the guidance scaling parameter, denoted with $\gamma$. To analyse this behaviour, we run the DA experiment on the KS dataset by using different scaling parameters. From Fig. 18 we can see that the value of $\gamma$ significantly influences the results, with the most significant differences seen when performing AAO sampling with 0 corrections. In this

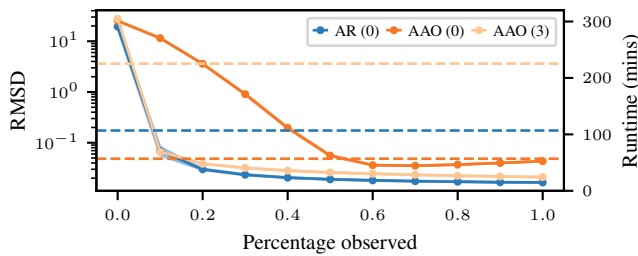

Figure 17: RMSD ($\downarrow$) computed on the KS dataset when varying the percentage of values observed. The value in bracket indicates the number of correction steps. Means and confidence intervals are estimated over 50 samples. Dashed lines represent the runtime for each method.

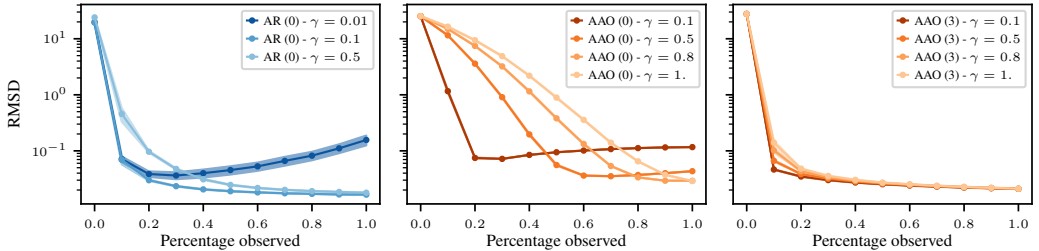

Figure 18: Ablation study on the effect of the guidance scaling parameter $\gamma$ in the data assimilation task. We report RMSD computed on the KS dataset when varying the percentage of values observed at each physical timestep for several different values of $\gamma$. We consider both the AR and AAO (with and without corrections) sampling procedures. Means and confidence intervals are estimated over 50 samples.

setting, we found that using corrections was crucial for making the AAO approach more robust (i.e. less sensitive to the value of $\gamma$).

The AR sampling scheme also seems to be impacted by the choice of $\gamma$. In particular, when the percentage of observed values is low (up to 20%), a smaller $\gamma$ (higher guidance strength) seems to be preferred. In contrast, when a lot of observations are available, if the guidance strength is too high (i.e. $\gamma$ is too small), the results get significantly worse. Therefore, it is clear that the best approach would be to fine-tune the guidance scaling parameter for each percentage of observed values. This opens up the possibility to investigate heuristics to automatically tune the scaling parameter depending on the number of observed values, but we leave this as future work.

## G    AUTOREGRESSIVE GUIDED SAMPLING WITH VARYING PREDICTIVE HORIZONS

As already eluded to in App. F, the autoregressive sampling method offers some flexibility in terms of the number of states generated at each iteration ($H$). From a computational budget perspective, the higher $H$, the fewer iterations need to be performed. However, this might come at the cost of a decrease in the quality of the generated samples. In this section, we investigate this trade-off for the Burgers' and KS datasets.

All results are shown for a window of 9, with varying predictive horizons $H$: 1, 2, 3, 4, 5, 6, 7, and 8. For each setting, the number of previously generated states we condition upon is $C = W - H = 9 - H$: 8, 7, 6, 5, 4, 3, 2, and 1. The trajectories are conditioned upon the first $9 - H$ true states.

The results are presented in terms of the MSE and correlation (Fig. 19), which are computed based on 200 test trajectories for the Burgers dataset, and on 128 test trajectories for the KS dataset. The number of diffusion steps used is 128. For all experiments, we used the DPM++ solver, with a linear time discretisation scheme. At the end, we also performed an additional denoising step, and adopted the dynamic thresholding method for the Burgers' dataset.

Fig. 20 also shows the RMSD for all the different studied settings for Burgers'. The performance is not very sensitive to $H$ for low to medium values of $H$, although the model seems to perform

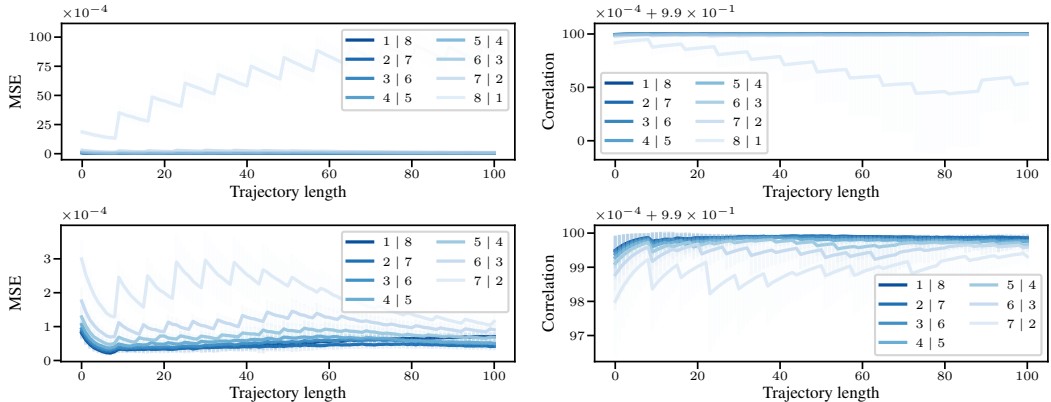

Figure 19: Experimental results on the Burgers' equation. Evolution of the MSE (left) and correlation (right) along the trajectory length. Bottom plots are a zoomed in version of the top ones, where we excluded the $8 \mid 1$ setting. The error bars indicate $\pm 3$ standard errors. Each line corresponds to a different predictive horizon $H$. Lower values of $H$ tend to lead to better performance metrics.

better with lower $H$ (and consequently, higher number of conditioning frames $C$). However, the performance quickly deteriorates for very high values of $H$ (i.e., $H = 8$).

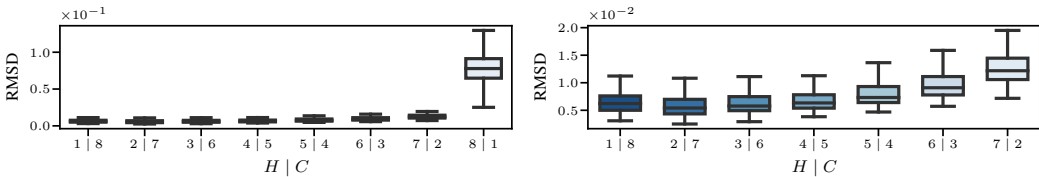

Figure 20: Burgers' dataset. RMSD for different values of $H$. The right plot is a zoomed in version of the left one, where we excluded the $8 \mid 1$ setting. Outliers have been removed.

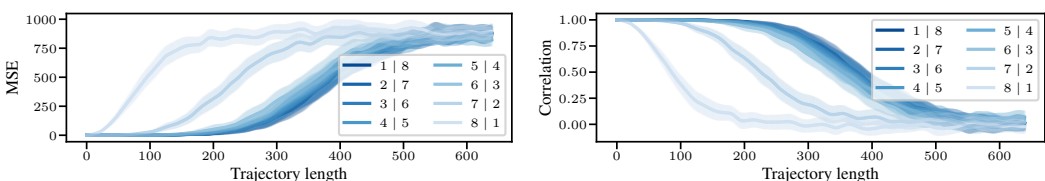

Figure 21: Experimental results on the Kuramoto-Sivashinsky equation. Evolution of the MSE (Left) and correlation (Right) along the generated trajectory length. The error bars indicate $\pm 3$ standard errors. Each line corresponds to a different predictive horizon $H$. The same conclusions as in the Burgers' experiment can be drawn.

Repeating the same experiment for the KS dataset, we see in Fig. 22 two summary metrics: RMSD and the high correlation time, defined as the time until the correlation drops to below 0.8. In general, lower RMSD and larger high correlation times are associated with better performance. The conclusions from the Burgers' dataset are confirmed for KS as well, with the best performing setting being $1 \mid 8$. However, the median RMSD does not decrease drastically even for higher $H$ (up to about $H = 6$).

## H COMPARISON BETWEEN GUIDED AND AMORTISED MODEL

In this section we analyse the following two conditioning approaches in the context of forecasting

- by directly learning a conditional model with a score network that is *amortised* over observations $y$ - *amortised model*.

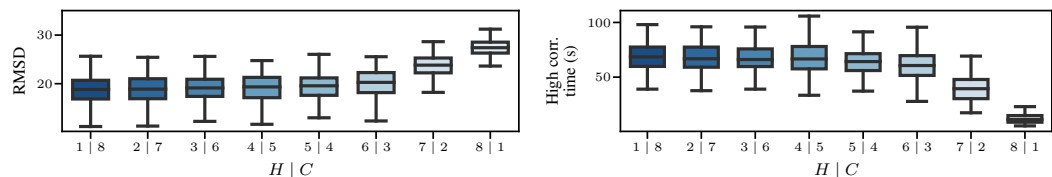

Figure 22: Experimental results on the Kuramoto-Sivashinsky equation. RMSD (left) and high correlation time (right) for different values of $H$. For RMSD, the smaller, the better, while the reverse is true for the high correlation time. Outliers have been removed.

- by learning a diffusion model over the prior, and sampling from the conditional *a posteriori*. Conditioning is introduced through reconstruction guidance, hence we will call this model the *guided model*.

We analyse two datasets: Burgers' and KS. For each dataset, the sampling in the guided model is performed autoregressively. We set the predictive horizon and the number of conditioning frames in such a way as to allow for a fair comparison between the amortised and the guided models.

## H.1 BURGERS' DATASET

In this case we study the following setting:

- Guided model with a window of 5, a prediction horizon $H$ of 2 and 3 conditioned frames. Amortised model, where we generate 2 samples at a time conditioned on the previous 3.

For the guided model, the conditioning from the first iteration uses a noisy version of the first three states (with standard deviation $\sigma_y = 0.01$), while for the amortised model, we use the first three true states. In the sampling process we use 128 diffusion steps. For the discretisation scheme, we study both the EI scheme (Zhang & Chen, 2023), as well as DPM++ solver with order 1 (Lu et al., 2023). The main difference between the two is that the first uses the $\epsilon$ prediction model, whereas the DPM++ solver uses the data prediction model. This allows us to also perform dynamic thresholding during the sampling process, which was shown to stabilise sampling with large guidance scales in the case of images (Ho et al., 2022a). Moreover, we can optionally take a final denoising step, specified through the parameter `denoise_to_zero`. Finally, we study the two time discretisation schemes introduced in App. E: quadratic and linear. The results are determined based on 200 test trajectories.

- **Final denoising step**: This was only studied for the DPM++ solver. The performance is clearly improved by using an extra denoising step at the end, as shown in Fig. 23. This is true for all models. In the amortised case, using the final denoising step helps avoid the jump in MSE around the initial states.

- **Solver and time discretisation**: The choice of solver and time discretisation schedule does not affect the results drastically, but it does lead to some differences. In the Burgers' dataset, the best setting seems to be to use the DPM++ solver with the linear time discretisation schedule when using 128 diffusion steps. However, this might depend on the dataset and on the number of diffusion steps used.

- **Guided vs. amortised**: Fig. 23 clearly shows the that guided model outperforms the amortised one both in terms of MSE (lower MSE), as well as correlation (higher correlation). The amortised model is better at modelling the initial states, as it does not contain any observation noise (unlike the guided one). However, errors in modelling initial states propagate and make the amortised model diverge faster than the guided one. Thus, the amortised model suffers from the main drawbacks of autoregressive models - because the conditioning is done through previously generated states, errors propagate rapidly and lead to divergent states. This is true in the guided model as well, but to a lesser extent. Moreover, the guided model has two components: the prior joint score and the conditional information from reconstruction guidance. It is possible that the influence from the prior model also helps to avoid divergence.

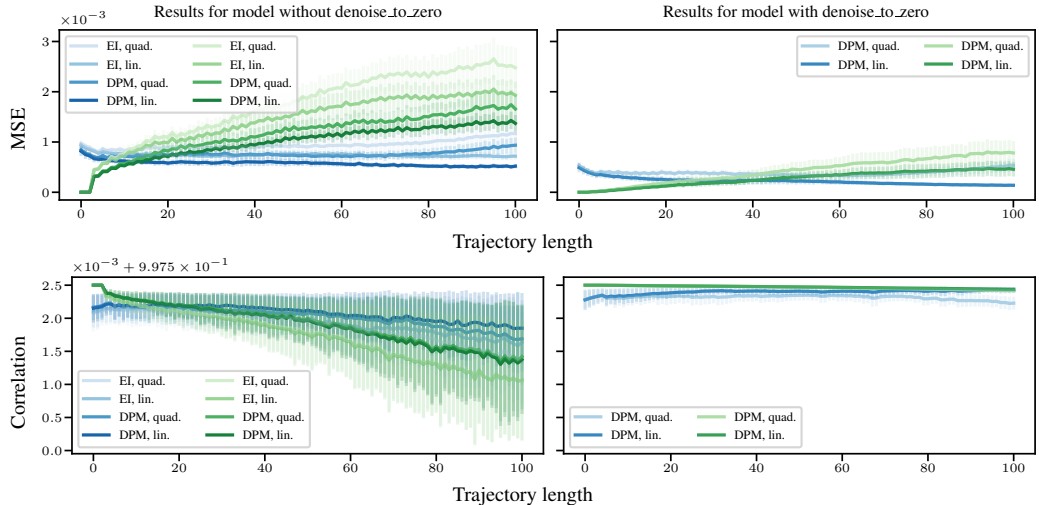

Figure 23: Evolution of the MSE (top) and Pearson correlation (bottom) along the trajectory length for the guided (blue) and amortised (green) models. In each model, we generate 2 states at a time conditioned on the previous 3. For each model, we study the EI and DPM++ solvers, alongside the quadratic and linear time discretisation. The left plot shows the results when we do not employ a final denoising step, while the right plot also performs one final denoising step. The final denoising step clearly improves performance in this setting, allowing the amortised model to avoid the jump in MSE around the first states. In the guided model, the initial error is dominated by the observation noise.

A summary of these findings can be found in Fig. 24, where we can easily see the positive impact of the final denoising step. Moreover, Fig. 24 shows that the guided model outperforms or performs on par with the amortised model in terms of RMSD. However, note that we only looked at the setting 2 | 3 in this analysis and the findings might not hold for other settings.

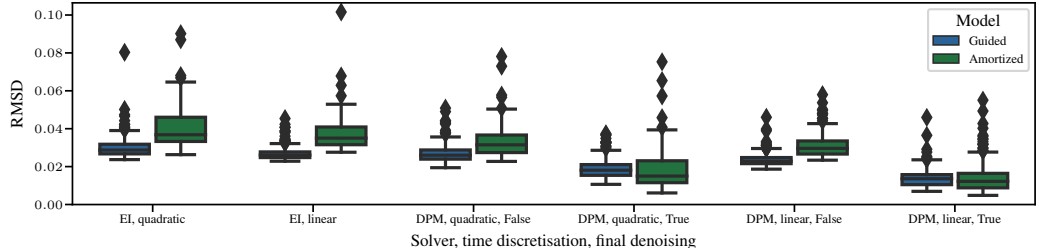

Figure 24: Comparison for Burgers' of RMSD for guided vs amortised model with different solvers and time schedules. For the DPM solver, we also show the results with and without taking a final denoising step. In this case we generate 2 states at a time, conditioned on the previous 3.

## H.2 KS DATASET

In this case we study the following settings:

- Guided model with a window of 5, a prediction horizon $H$ of 2 and 3 conditioned frames. Amortised model, where we generate 2 samples at a time conditioned on the previous 3.

- Guided model with a window of 9, a prediction horizon $H$ of 3 and 6 conditioned frames. Amortised model, where we generate 3 samples at a time conditioned on the previous 6.

- Varying window sizes ($H$ and $C$) for guided and amortised models.

For the KS dataset we did not use dynamic thresholding, but still used the final denoising step. We also just investigated the DPM++ solver with either the quadratic or linear time schedule. The results are derived on 128 test trajectories, for a trajectory length of 640 time steps, corresponding to 128s.

As a summary metric, besides the RMSD, we also compute the high correlation time, defined as the time until the correlation drops to below 0.8 (the higher, the better).

### H.2.1 GENERATE 2 CONDITIONED ON 3

Fig. 25 shows that the guided model performs better than the amortised one on average, although the metrics of both models are within error bars. However, we only analysed the setting 2 | 3 and the findings might not hold for other settings. In particular, we observed that the guided model tends to perform better with more conditioning information, whereas increasing $C$ harms the performance of the amortised model. The summary metrics over the entire trajectory are shown in Fig. 26. The

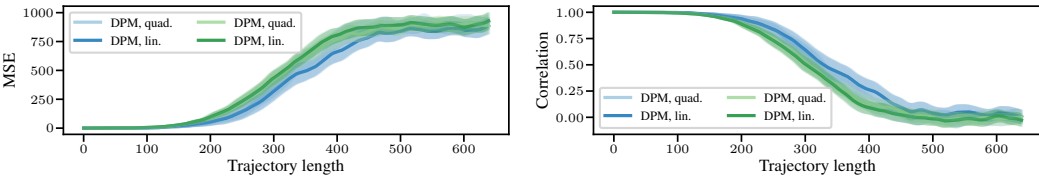

Figure 25: Evolution of the MSE (left) and Pearson correlation (right) along the trajectory length for the guided (blue) and amortised (green) model for the KS dataset. In this case, we generate 2 states conditioned on 3 (with a window of 5 for the guided model). Two different time schedules are investigated: quadratic and linear.

plots show that the guided model has better median metrics, but the two models are comparable when taking into account the error bars.

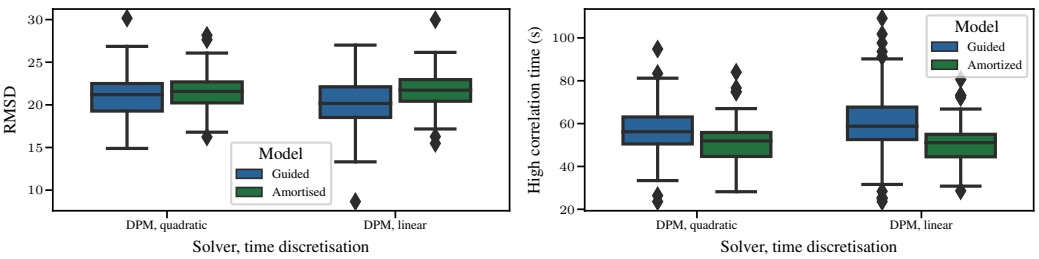

Figure 26: RMSD (left) and high correlation time (right) for the guided and amortised model for the KS dataset. In this case, we generate 2 states conditioned on 3 (with a window of 5 for the guided model). For RMSD, the lower the value, the better the model, while for the high correlation time a larger value is preferred.

### H.2.2 GENERATE 3 CONDITIONED ON 6

Similar conclusions as in the previous section can be drawn if we generate 3 states at a time conditioned on the previous 6. Fig. 28 suggests that the difference between the two models is more pronounced in this setting, with the guided model outperforming the amortised one.

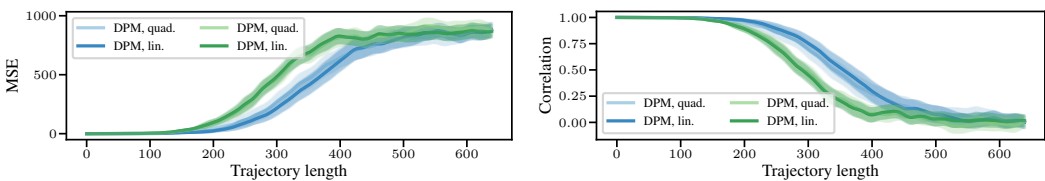

Figure 27: Evolution of the MSE (left) and Pearson correlation (right) along the trajectory length for the guided (blue) and amortised (green) model. In this case, we generate 3 states conditioned on 6 (with a window of 9 for the guided model).

### H.2.3 VARYING WINDOW SIZE

Fig. 29 shows the performance of the amortised model for different $H \in \{1, 2, 4, 6, 8, 12\}$ and $C \in \{1, 2, 4, 8\}$. Regardless of $C$, the model underperforms for small $H = 1$, with high correlation

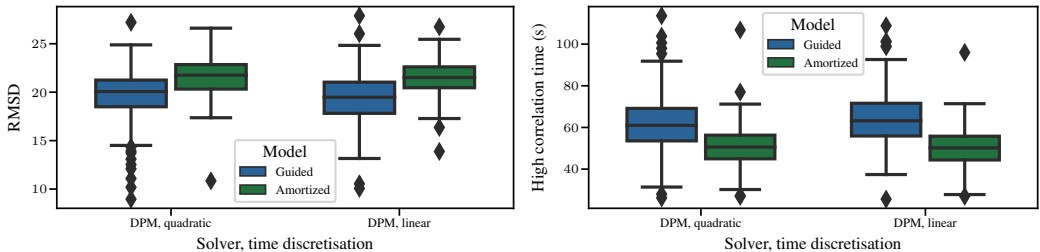

Figure 28: RMSD (left) and high correlation time (right) for the guided and amortised model when generating 3 states and conditioning on 6 for KS. The difference between the guided and the amortised model is more pronounced than in the case when we generate 2 states conditioned on 3 (see Fig. 26).

time of less than $50$s. We hypothesise that the reason for this might be the combination of the following: i) generating larger number of frames at the same time requires less steps for the same trajectory length, so there is less error accumulation; and ii) for larger $H$ the model is trained with a better learning task compared to $H = 1$, as larger $H$ provides more information to the model. For $H \geq 4$, the model performs similarly for any $H$, being slightly more sensitive to larger $C$ when $H$ is smaller. Regardless of $H$, the performance of the model deteriorates significantly with larger $C = 8$, similar to the behaviour shown in Lippe et al. (2023).

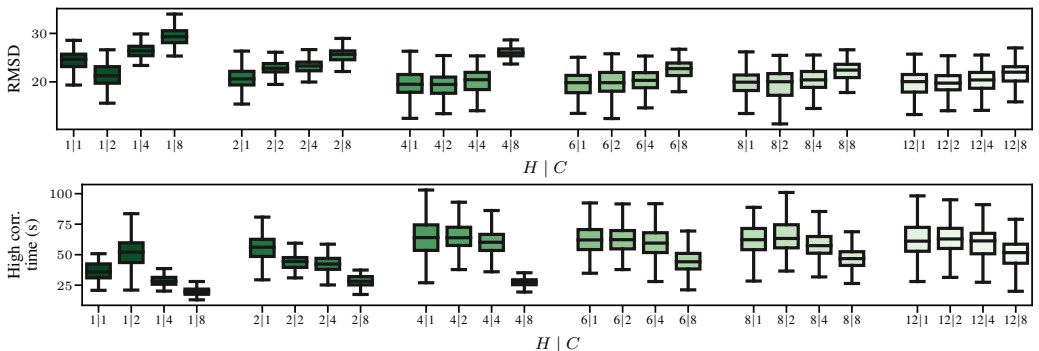

Figure 29: RMSD (top) and high correlation time (bottom) for the amortised model with varying $H$ and $C$. The model performs the best when generating relatively large blocks ($H \geq 4$) and conditioning on the smaller number of frames ($C \leq 4$).

In contrast, the performance of the guided model is not that sensitive to the window size, as shown in Fig. 30. The best performance is observed for medium window sizes ($W = 9$); if the window is too small, the effective Markov order assumed might be too low. As the window increases, so does the effective order, but there is probably a limit to how much increasing the effective Markov order can help performance. While the noised up states within the blanket might have infinite Markov order, the correlation between them likely shows a sharp decrease, so there is little gain between using a window of 9 or a window of 11. Moreover, increasing the window size might make the optimisation harder, explaining why we don't see improvement in performance beyond $W = 9$.

**Conclusion on guided vs. amortised model**   The investigation showed that the guided and amortised model are governed by different dynamics - the guided model generally seems to benefit from more conditioning information (high $C$) and from generating only a few states at a time (low $H$). In contrast, the amortised model performs best when generating a larger number of frames at a time (high $H$) and conditioning on fewer frames (low $C$). This means that the best $H \mid C$ setting for each model will likely be different. However, the guided model has the benefit that it can be queried in any setting, without any further training. In contrast, to discover the best setting in the amortised case, a model needs to be trained for each $H \mid C$ combination.

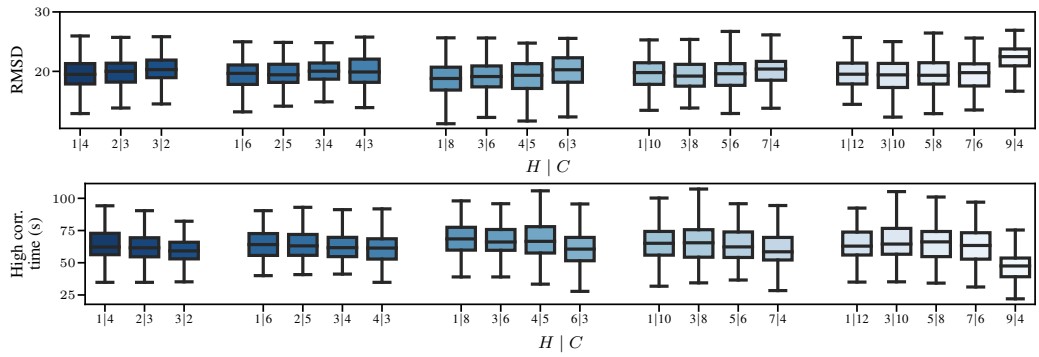

Figure 30: RMSD (top) and high correlation time (bottom) for the guided model with varying window sizes (5, 7, 9, 11, 13) and different $H \mid C$ combinations. The model is not highly sensitive to window size, with best median performance seen for medium window size ($W = 9$) in the setting $1 \mid 8$.

## I GUIDANCE SCHEDULE

Sec. 4 introduces the four types of guidance studied in this paper. This section provides a comparison between these guidance formulations, applied on the Burgers' and KS datasets in the context of forecasting. The results are derived using AR sampling, using models with a window of 9, with $H = 1$ and $C = 8$ ($1 \mid 8$). Additionally, we also condition on the first initial 8 states. In all experiments from this section 128 diffusion steps were employed, and the solver used was DPM++ with a linear time discretisation schedule. At the end, one final denoising step was taken. For both datasets, we also studied the influence of dynamic thresholding, focusing on the stability of the methods at high guidance strenghts.

### I.1 BURGERS' EQUATION

We analysed several settings of the $\gamma$ hyperparameter: [0.01, 0.03, 0.05, 0.1, 0.5, 1] for SDA, [0.001, 0.01, 0.03, 0.1, 1] for DPS, [0.0002, 0.001, 0.01, 0.2, 1.0, 10.0] for VideoDiff, and [0.02, 0.1, 0.2, 2.0, 10.0, 100.0] for ΠGDM. Note that we have studied different $\gamma$ ranges - for each guidance mechanism the object $\gamma$ is multiplied by has different scales, so the most appropriate $\gamma$ values will vary from one mechanism to another.

The results in Fig. 31 correspond to the best setting for each guidance type, indicating that in terms of RMSD, SDA has the best performance, followed by ΠGDM, DPS, and then Video diffusion.

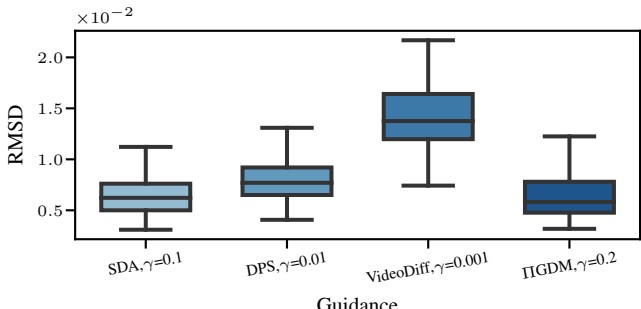

Figure 31: RMSD for the Burgers' dataset using different types of guidance (and the best corresponding $\gamma$ value). Outliers have been removed. SDA and ΠGDM show the best results, with a similar performance in terms of RMSD, followed by DPS and then Video diffusion.

**Stability of each guidance type**  Fig. 32 shows the RMSD for each guidance type when applied to the Burgers' data, both with and without using dynamic thresholding.

By comparing the four plots we can see that

- The lowest RMSD is achieved when using dynamic thresholding for all four types of guidance. However, SDA is the only guidance mechanism that is able to achieve similar performance (within error bounds) without the use of dynamic thresholding. For all the other three guidance types, not using dynamic thresholding resulted in a significant increase in RMSD for the best $\gamma$ setting.

- The extent of the increase in RMSD resulting from not employing dynamic thresholding (i.e., lowest RMSD achieved with vs without dynamic thresholding) varies across guidance mechanisms

  - SDA: negligible increase;
  - DPS: significant, but relatively small increase; Averaged across all test samples, the mean RMSD increased more than five times (from $0.8 \times 10^{-2}$ to $4.5 \times 10^{-2}$);
  - Video diffusion: big increase in RMSD, with the average across the 200 test samples increasing by almost two orders of magnitude;
  - $\Pi$GDM: big increase in RMSD, with the average across the 200 test samples increasing by more than two orders of magnitude.

- For the investigated guidance strengths, DPS is the only guidance mechanism that did not lead to numerical overflow. For high guidance strengths the results were highly inaccurate, but not unstable. We hypothesise this is due to the $\|y - A\hat{x}(x(t))\|$ term that probably promotes stability.

These findings suggest that while the SDA guidance leads to similar results regardless of whether other tricks are employed (such as dynamic thresholding), the performance of other guidance types is more sensitive to them. In particular, dynamic thresholding was introduced to stabilise sampling at high guidance strengths (low $\gamma$), and it seems to be crucial for all guidance types but SDA.

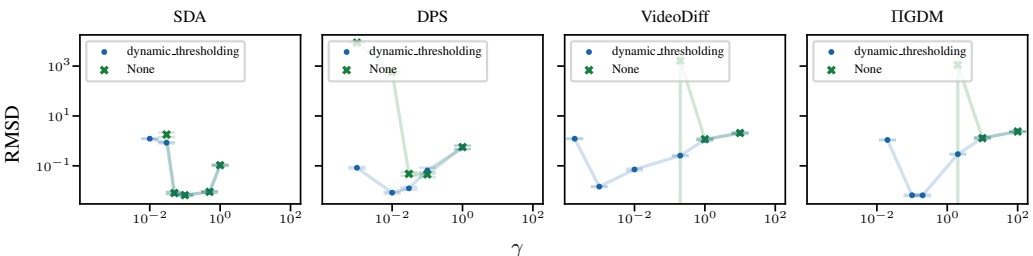

Figure 32: RMSD for the four types of guidance using different values of the $\gamma$ hyperparameter for Burgers'. Note that a lower $\gamma$ implies a higher guidance strength. Also note that the scale of the x- and y-axes is logarithmic. For each $\gamma$, if the plot does not show a result for the case in which dynamic thresholding was not employed (green X), this is because the results were unstable.

Fig. 33 also shows some example test trajectories and the corresponding sampled trajectories for the best setting of each guidance type. For each guidance mechanism, we also show the absolute error between the test samples and the generated trajectories.

### I.2 KS EQUATION

#### I.2.1 MODEL USING ORIGINAL DATA

The models used in Sec. 4 were trained on the original KS data (i.e., no pre-processing was applied). Given that the training data values lie in the range [-3.61, 3.56], dynamic thresholding could not be applied for this model. One requirement for dynamic threhsolding to work is for the data values to lie within [-1, 1]. Thus, we investigate the performance of each guidance type without using dynamic thresholding.

Fig. 34 shows that SDA has a significantly better performance as compared to all other guidance types. This is in line with the findings from the Burgers' dataset, where we found dynamic thresholding to be crucial for DPS, Video diffusion, and $\Pi$GDM to achieve a similar performance to SDA.

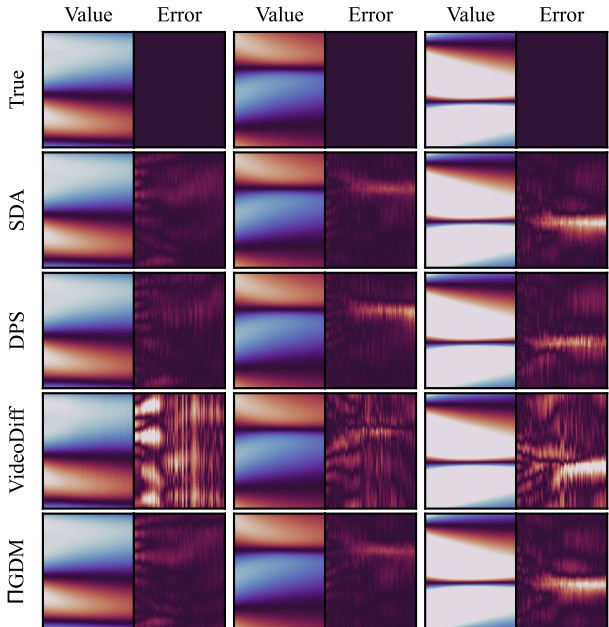

Figure 33: True test trajectories (top), followed by trajectories generated using SDA, DPS, Video diffusion, and ΠGDM for the Burgers' dataset. The results are shown for three example test trajectories. For each guidance mechanism, near the generated samples, we also show the absolute error between them and the ground truth. The scale is the same between the error images.

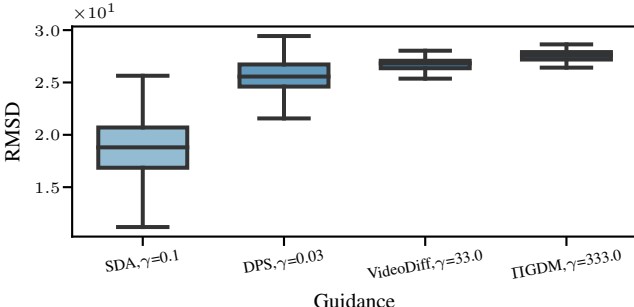

Figure 34: RMSD for the KS dataset using different types of guidance (and the best corresponding $\gamma$ value). Outliers have been removed. SDA has the best performance by far, with the other guidance types achieving much higher RMSD when dynamic thresholding is not employed.

### I.2.2  MODEL USING TRANSFORMED DATA

To confirm the findings from the Burgers' dataset, we also investigated the four guidance types with and without dynamic thresholding on a model trained on transformed KS data; we modified the data range to [-1, 1], such that dynamic thresholding could be employed. The following $\gamma$ ranges were studied: [0.03, 0.05, 0.1, 0.5, 1.0] for SDA, [0.001, 0.005, 0.01, 0.05, 0.1, 1.0] for DPS, [0.0003, 0.001, 0.002, 0.01, 0.1, 0.5] for VideoDiff, and [0.02, 0.1, 0.2, 2.0, 10.0, 100.0] for ΠGDM.

Fig. 35 confirms that dynamic thresholding is crucial for stable sampling at high guidance strengths (low $\gamma$) for DPS, Video diffusion, and ΠGDM, allowing them to reach a performance similar to SDA. In contrast, SDA is not that sensitive to dynamic thresholding - the RMSDs with and without dynamic thresholding are not significantly different to one another in the best setting ($\gamma = 0.1$).

Moreover, we found that the best setting of $\gamma$ was consistent between datasets for SDA and DPS. ΠGDM showed a similar behaviour - for both datasets, the results for $\gamma = 0.1$ and $\gamma = 0.2$ were

among the best two, and not significantly different (i.e., within error bounds). Video diffusion was the only guidance type where the best $\gamma$ setting was dataset-dependent.

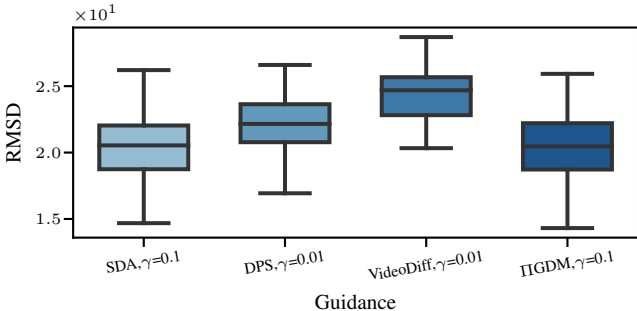

Figure 35: RMSD for the transformed KS dataset using different types of guidance (and the best corresponding $\gamma$ value). Outliers have been removed. SDA and ΠGDM have a similar performance, followed by DPS and Video diffusion.

The same conclusion can be drawn from Fig. 36, where we show that employing dynamic thresholding does not improve the RMSD significantly for SDA, but it has a crucial effect for the other three guidance mechanisms.

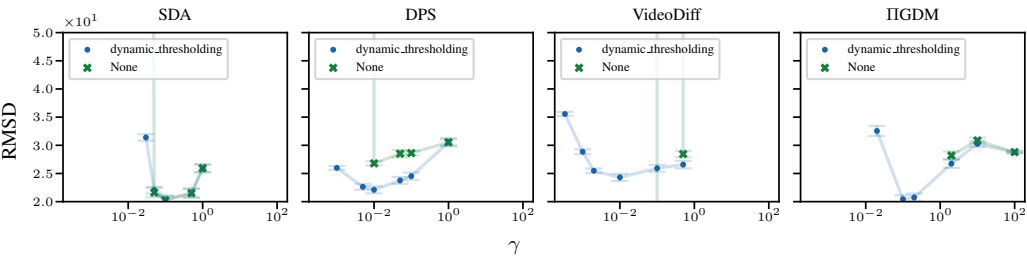

Figure 36: RMSD for the four types of guidance using different values of the $\gamma$ hyperparameter for the transformed KS dataset. Note that a lower $\gamma$ implies a higher guidance strength. Only values between 20-50 are shown on the y-axis for clarity. For each $\gamma$, if the plot does not show a result for the case in which dynamic thresholding was not employed (green X), this is because the results were unstable.

Finally, one test trajectory for the best setting of each guidance mechanism is shown in Fig. 37.

## J  SENSITIVITY TO NOISE IN OBSERVATIONS

In this section we study the robustness of the guided AR and amortised AR models to noise in the initial observations in the context of forecasting. More specifically, for each model we analyse its forecasting performance (in terms of high correlation time) in the following two settings

1. Condition on the true initial observations.
2. Condition on noised initial observations, with noise with standard deviation $\sigma_y = 0.01$.

The models we analyse are the same as the ones in Sec. 4. For the amortised case, we study both a model that was trained on noiseless conditional frames ($s = 0$), as well as one that was trained on noisy frames with fixed noise $s = 0.01$, with the hypothesis being that the latter is more robust to noise in the initial observations.

1. Guided AR model, generating 1 state at a time and conditioning on 8 (1|8). We also condition on the initial 8 true or noised up states.
2. Amortised AR models, generating 7 states at a time and conditioning on 2 (7|2). We also condition on the initial 2 true or noised up states.
   - Trained on noiseless conditional frames $s = 0$

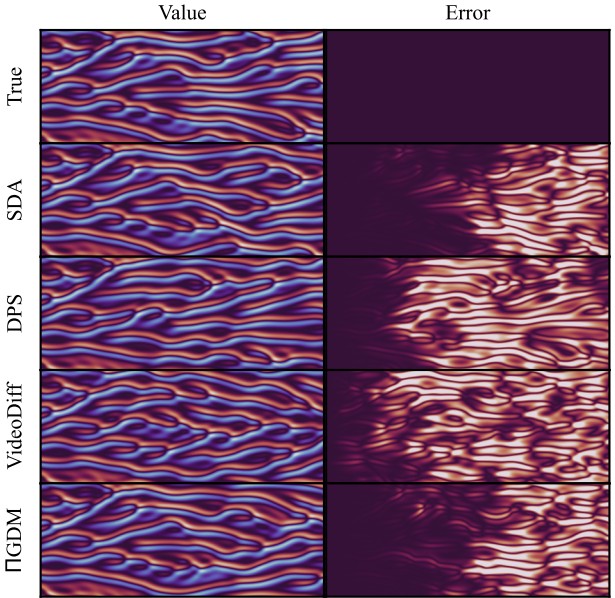

Figure 37: True test trajectory (top), followed by trajectories generated using SDA, DPS, Video diffusion, and ΠGDM for the KS dataset. For each guidance mechanism, near the generated samples, we also show the absolute error between them and the ground truth. The scale is the same between the error images.

- Trained on noisy conditional frames $s = 0.01$

We used the DPM++ solver with a linear time discretisation schedule for all models from this section. We performed 128 diffusion steps with no Langevin corrector steps, and performed a final denoising step, as discussed in Sec. 4.

The results on the KS dataset are shown in Fig. 38. We can see that the guided model (using the SDA guiding mechanism) is very robust to noise in the initial observations, showing only a slight decrease in high correlation time when we condition on noisy initial states, as opposed to the true ones (less than 2s). In comparison, the performance of the amortised model that was trained on noiseless conditional frames ($s = 0$) suffers more, with a decrease in high correlation time of more than 7s (see right-most green bars in Fig. 38). We can also equip the amortised model with a similar robustness to noise in the initial states by training it on noisy conditioned frames (see left-most green bars in Fig. 38). This strategy clearly leads to less sensitivity to the amount of noise present in the initial observations, with a decrease in high correlation time of less than 2s. However, although more robust to noise, it is significantly inferior to the original amortised model (trained on noiseless conditioned frames), managing to achieve a high correlation time of only about 50s.

Thus, it seems that the guided model is inherently less sensitive to the amount of noise present in the initial states. While there are strategies to make the amortised model similarly robust to noise, they significantly harm performance.

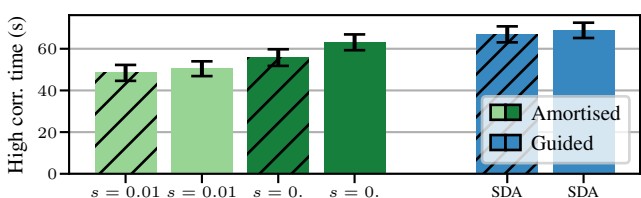

Figure 38: High correlation time for the KS dataset for the amortised (green) and guided (blue) models. The hatched bars indicate that the initial observations we conditioned upon were noisy (with $\sigma_y = 0.01$), whereas the plain ones indicate that we conditioned on the true initial states. For the amortised models, the left-most two bars correspond to the model trained with noise ($s = 0.01$), whereas the other two bars correspond to the model trained without noise ($s = 0$).

