# OpenReview forum: "Guided Autoregressive Diffusion Models with Applications to PDE Simulation"
_ICLR.cc/2024/Workshop/AI4DiffEqtnsInSci — AI4DiffEqtnsInSci @ ICLR 2024 Poster_

### Official Review · Reviewer_5NLX · 2024-02-26
**Exhaustive evaluation of a marginally superior method**

**Rating:** 7
**Confidence:** 4

**Review:**

### Summary:
A guided autoregressive diffusion model is introduced to forecast and assimilate PDE dynamics. The model makes use of conditional frames that provide context of the problem and can be trained on short slices of long sequences.

### Strengths:
- Very appealing to see that GuARD benefits from more context frames that are provided (i.e., using higher Markov orders). This actually matches people's intuition, but seems not to be the case for Amortized.
- Detailed information in the appendix (I did not check it thoroughly)

### Weaknesses
- Despite the concrete second paragraph of the introduction, I could not quite extract the contribution and novelty of this work. Can you elaborate more specifically what is new and in what way?
- GuARD only marginally improves over the standard Amortized technique. Nevertheless, GuARD is faster and slightly better.

### Questions:
1. Is $x(0)$ the noise-free state? Maybe make more explicit in the **Continuous-time diffusion models** paragraph.
2. How it the density $p_t$ of $x(t)$ calculated, or what does it concretely represent?
3. How do Langevin Monte Carlo (corrector) steps look like?
4. In Figure 2, bottom right, why is the Amortized AR model worse if conditioning on many frames an producing a single frame (1|8) than when conditioning on a single frame and producing many frames (8|1)? At the same time, why is GuARD worse than Amortized in the (7|2) configuration?

### Minor comments:
- Equal sign missing before $\Pi$ in second line of page 3?

---

### Official Review · Reviewer_jgVu · 2024-02-26
**The authors present interesting experiments and results, however due to the denseness of the content several aspects remain unclear.**

**Rating:** 6
**Confidence:** 3

**Review:**

The paper introduces GUARD, a diffusion model which is trained on short PDE trajectories and aims to forecast longer trajectories and perform data assimilation given sparse spatial data.

Pros
- The approach and comparisons are interesting, experiments are well-described.
- The results presented seem promising (especially for forecasting) and should be investigated further.


Quality: The quality of the paper is good. \
Clarity: While the paper is written clearly it sometimes lacks in logic as crucial parts are omitted/in the appendix. \
Originality: The basis of GUARD is AR (Ho et al.) and the distinction to this work remains unclear. Due to a lack of conclusion it is also not clear what exactly the authors claim to be new.  \
Significance of this work: The results presented are promising. However, to determine the significance a more elaborate benchmark would be needed.

Major issues
- The paper lacks a discussion and conclusion. Overall the background section and description of the setup take up most space of the paper and sections on the experiments performed and their results remain very short. Therefore, many things seem to be omitted and the appendix is referenced often.

- The experiments performed for data assimilation and forecasting are not comparable to each other. It remains unclear why for data assimilation only the two approaches AAO and AR are compared while for forecasting GUARD is compared to an amortized model. Also, the amortized model is not described and there's a lack of further benchmarks.


Minor issues
- Math equations are difficult to read when inline.
- Error metrics: Why are some evaluations performed using MSE and others using RMSD? 2a vs 2c.
- Runtime comparison of GUARD and amortized model?
- Conditioned on less states the amortized model performs better than GUARD (Figure 2). Why? Could this be a reason to use the amortized model in some cases and GUARD in others?

---

### Meta-Review · Area_Chair_hcTj · 2024-02-29

**Recommendation:** Accept (Poster)

**Metareview:**

This paper presents  the Guided AutoRegressive Diffusion model (GuARD), which is able to generate accurate predictions of long PDE trajectories. The concerns of clarification raised by both reviewers are valid. Those concerns can be addressed in the camera ready version.

---

### Decision · Program_Chairs · 2024-02-29

Accept (Poster)